# Differential requirements for Smarca5 expression during hematopoietic stem cell commitment
Tereza Turkova [1,4], Juraj Kokavec [1,4], Tomas Zikmund [1], Nikol Dibus [2], Kristyna Pimkova [1], Dusan Nemec [1], Marketa Holeckova[1], Livia Ruskova[1], Radislav Sedlacek [3], Lukas Cermak [2]✉ & Tomas Stopka [1]✉

The formation of hematopoietic cells relies on the chromatin remodeling activities of ISWI ATPase SMARCA5 (SNF2H) and its complexes. The *Smarca5* null and conditional alleles have been used to study its functions in embryonic and organ development in mice. These mouse model phenotypes vary from embryonic lethality of constitutive knockout to less severe phenotypes observed in tissue-specific *Smarca5* deletions, e.g., in the hematopoietic system. Here we show that, in a gene dosage-dependent manner, the hypomorphic allele of SMARCA5 (*S5^tg*) can rescue not only the developmental arrest in hematopoiesis in the hCD2iCre model but also the lethal phenotypes associated with constitutive *Smarca5* deletion or Vav1iCre-driven conditional knockout in hematopoietic progenitor cells. Interestingly, the latter model also provided evidence for the role of SMARCA5 expression level in hematopoietic stem cells, as the Vav1iCre *S5^tg* animals accumulate stem and progenitor cells. Furthermore, their hematopoietic stem cells exhibited impaired lymphoid lineage entry and differentiation. This observation contrasts with the myeloid lineage which is developing without significant disturbances. Our findings indicate that animals with low expression of SMARCA5 exhibit normal embryonic development with altered lymphoid entry within the hematopoietic stem cell compartment.

Cellular differentiation begins in the fertilized egg and is characterized by gradual and extensive reprogramming of gene expression, especially during gastrulation, which is dependent on changes in chromatin accessibility and activity. These changes are orchestrated by chromatin remodeling enzymes, which are essential for cell differentiation and organism development[1]. Their main role is to mechanically catalyze the process of nucleosome sliding, either in cooperation with transcription factors or independently as a result of their DNA binding activity[2]. Several families of chromatin remodeling complexes differ in the presence of catalytic and regulatory subunits, for example the Imitation Switch Complex (ISWI) family, which contains either SWI/SNF Related, Matrix Associated, Actin Dependent Regulator Of Chromatin, Subfamily A, Member 1 (SMARCA1) or SMARCA5 as a catalytic subunit[3]. ISWI complexes play broad and essential roles in nearly all genetic processes, including activation or repression of transcription, DNA repair, and DNA replication. They act either by

maintaining evenly spaced nucleosomal arrays or by repositioning nucleosomes near promoters to increase the accessibility of DNA[4].

Germline deletion of *Smarca5* leads to early embryonic lethality due to the arrest of embryonic stem cell (ESC) differentiation[5]. The knockout of *Smarca5* in ESCs directly demonstrated the role of SMARCA5 in the regulation of nucleosomal distance (linker length). Its absence led to an increased linker length, a defect that changed accessibility for many transcription factors, including the binding of a CCCTC binding factor (CTCF) to the imprinted DNA regions[6]. The main function of CTCF in these regions is the inhibition of transcriptional enhancers, which is one of mechanisms controlling the expression of the transcription factor PU.1 (SPI1). PU.1 is essential for the early initiation of hematopoiesis and its differentiation towards myelopoiesis and lymphopoiesis[7]. In zebrafish, SMARCA5 facilitates the binding of transcription factors that control the expression of hematopoietic regulators such as bcl11ab[8]. This process mediates the

[1]Hematology Laboratories, BIOCEV; 1st Faculty of Medicine, Charles University, Vestec, Czech Republic. [2]Laboratory of Cancer Biology, Institute of Molecular Genetics of the Czech Academy of Sciences, Prague, Czech Republic. [3]Czech Centre for Phenogenomics, Institute of Molecular Genetics of the Czech Academy of Sciences, Vestec, Czech Republic. [4]These authors contributed equally: Tereza Turkova, Juraj Kokavec. ✉e-mail: lukas.cermak@img.cas.cz; tstopka@lf1.cuni.cz

acquisition of definitive hematopoietic stem and progenitor cell (HSPC) characteristics and facilitates development into definitive fetal HSPCs. In addition, gene expression and chromatin accessibility data from zebrafish HSPCs suggest that SMARCA5 is involved in controlling the activity of genes associated with HSPC expansion and differentiation[8].

Using conditional inactivation of *Smarca5* in murine hematopoietic progenitors, we followed up on the HSPC study and demonstrated the essential role of SMARCA5 during early HSPC differentiation. Most importantly, in this model, we observed accumulation of hematopoietic progenitors and activation of p53 targets. These changes appear to disrupt normal erythropoiesis and promote premature death of animals during fetal development[9]. To prevent this phenomenon and to study in detail the role of SMARCA5 in early definitive hematopoiesis, we decided to create a transgenic model with a hypomorphic expression (lower expression on protein level) of SMARCA5 in the context of its endogenous gene deletion. This approach allowed us to introduce a model with a graded SMARCA5 expression. Using cell biology and bone marrow (BM) transplantation techniques, we have shown that SMARCA5 is involved in the very early differentiation of HSPCs, a process with significant consequences for terminal maturation, especially in the lymphoid lineage.

## Results

### Hypomorphic *hCD2-S5^tg* allele rescues the developmental thymocyte arrest in *Smarca5* null mutants

Since heterozygotes carrying a null allele of the *Smarca5* (*S5*) gene have normal hematopoiesis while homozygotes display severe embryonic defects, we created a model using a conditional hypomorphic allele of the *S5* gene (*S5^tg*, Fig. 1a) to study how different levels of SMARCA5 affect

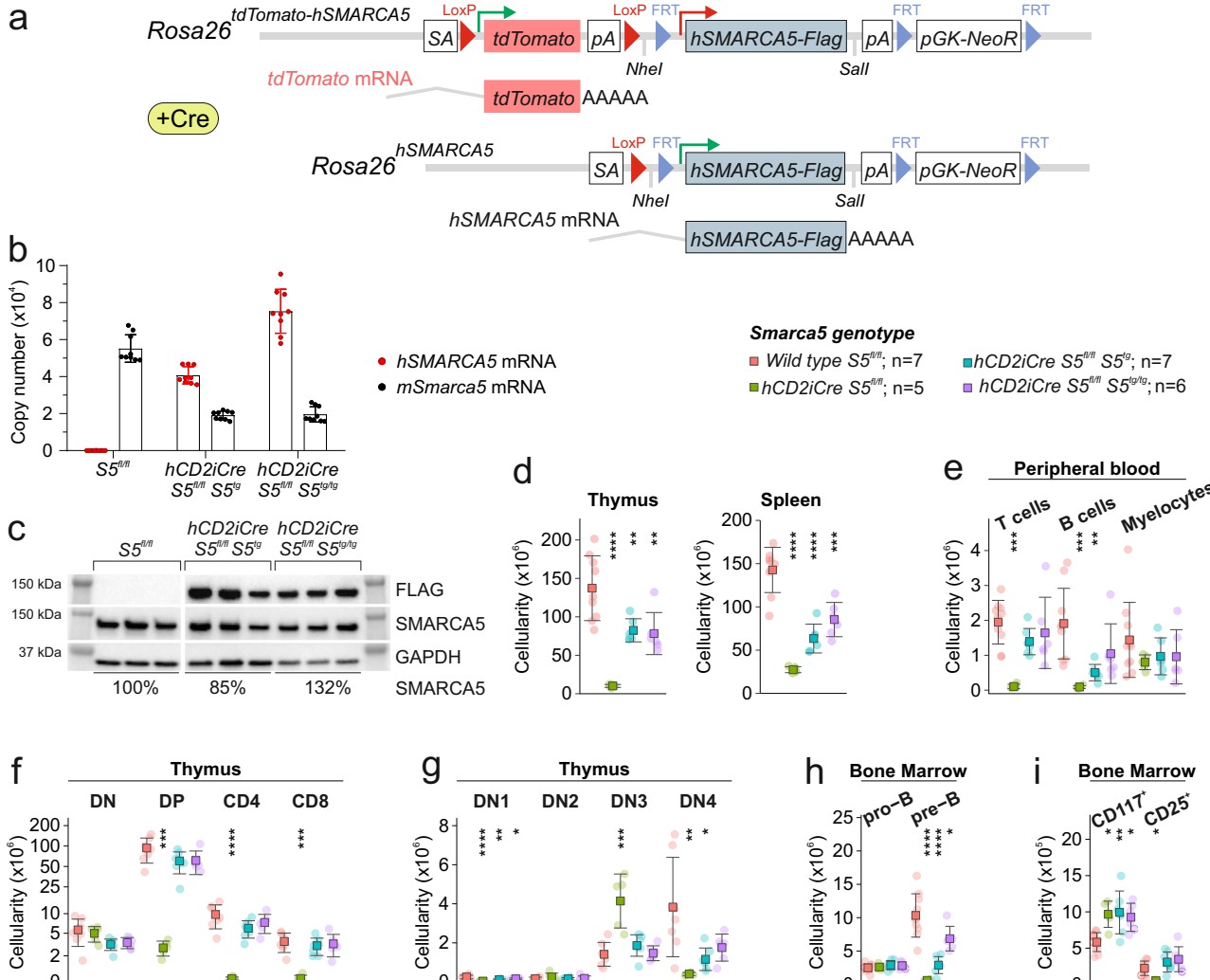

**Fig. 1 | Transgenic *SMARCA5* releases the blockade of T and B cell differentiation after *Smarca5* deletion. a** Scheme of recombination at the *Rosa26* locus: incorporation of a neomycin resistance construct (*pGK-NeoR*) containing the *tdTomato* gene with polyadenylation signal (pA) and *loxP* sites (red triangles) and *SMARCA5-FLAG* cDNA with FRT sites (blue triangles) downstream of the splice acceptor (SA) site. The transgene expresses *tdTomato* and, upon activation of Cre recombinase, the *tdTomato* cDNA is removed, leading to the expression of *SMARCA5*. **b** Copy number quantification of mouse and human *Smarca5* mRNA by qRT-PCR (AVG ± SD, *n* = 3) in thymi of 2-month-old mice of the indicated genotypes after normalization to *Gapdh* and *Hprt* expression. **c** Immunoblot of thymi from 8-week-old mice of the indicated genotypes stained with anti-FLAG or anti-SMARCA5 antibodies. GAPDH staining was used as a loading control. Signal density is stated in % relative to controls. **d** Thymic and splenic cellularity of 8-week-old mice of the indicated genotypes (analyzed by flow cytometry). **e** Cytometry with anti-CD4/CD8, anti-B220, and anti-Gr-1/CD11b from CD45[+] peripheral blood cells of the indicated genotypes. **f** Cytometric analysis of thymic populations: DN, DP, and SP of the indicated genotypes. **g** Cytometric analysis of thymocytes (negative for B220, Gr-1, CD11b, CD11c and Nk1.1) of the indicated genotypes using anti-CD25 and anti-CD44 antibodies. **h, i** Cytometric analysis of B cell development of the indicated genotypes using anti-B220 and anti-CD43 staining of BM CD45[+] cells (**h**) and anti-CD117 and anti-CD25 staining of pro-B cells (CD43[+]B220[+]) (**i**). Statistics: One-Way ANOVA with Tukey's Honestly Significant Difference test (*p* adjusted value: *p* < 0.05 = *, *p* < 0.01 = **, *p* < 0.001 = ***, *p* < 0.0001 = ****, no asterisks = non-significant), the error bars represent standard deviation.

**Table 1 | The table shows the numbers and % of mice born with the respective genotypes from the crosses of the parents described in the left part**

| *hCD2iCre S5*[tg] | Progeny genotypes | | | |
|---|---|---|---|---|
| **Parental genotypes** | *S5*[fl/fl] *S5*[tg] | *S5*[fl/fl] *S5*[tg/tg] | *hCD2iCreS5*[fl/fl] *S5*[tg] | *hCD2iCre S5*[fl/fl] *S5*[tg/tg] |
| ♂ *hCD2iCre S5*[fl/fl] *S5*[tg] x ♀ *S5*[fl/fl] *S5*[tg/tg] | 141 (24.6%) | 151 (26.3%) | 140 (24.4%) | 142 (24.7%) |

| *Vav1iCre S5*[tg] | Progeny genotypes | | | |
|---|---|---|---|---|
| **Parental genotypes** | *S5*[fl/fl] *S5*[tg] | *S5*[fl/fl] *S5*[tg/tg] | *Vav1iCre S5*[fl/fl] *S5*[tg] | *Vav1iCre S5*[fl/fl] *S5*[tg/tg] |
| ♂ *Vav1iCre S5*[fl/fl] *S5*[tg] x ♀ *S5*[fl/fl] *S5*[tg/tg] | 101 (24.3%) | 104 (25.0%) | 94 (22.6%) | 117 (28.1%) |

| *ActB-Cre S5*[tg] | Progeny genotypes | | |
|---|---|---|---|
| **Parental genotypes** | *S5*[wt/wt] *S5*[tg/tg] | *S5*[del/wt] *S5*[tg/tg] | *S5*[del/del] *S5*[tg/tg] |
| **Prenatal E13.5** | | | |
| ♂ *S5*[del/wt] *S5*[tg/tg] x ♀ *S5*[del/wt] *S5*[tg/tg] | 15 (34.9%) | 26 (60.5%) | 2 (4.7%) |
| ♂ *S5*[fl/fl] *S5*[tg/tg] x ♀ *ActB-Cre S5*[del/wt] *S5*[tg] | 0 | 21 (95.5%) | 1 (4.5%) |
| **Prenatal E18.5** | | | |
| ♂ *S5*[del/wt] *S5*[tg/tg] x ♀ *S5*[del/wt] *S5*[tg/tg] | 21 (36.2%) | 33 (56.9%) | 4 (6.9%) |
| **Postnatal** | | | |
| ♂ *S5*[del/wt] *S5*[tg/tg] x ♀ *S5*[del/wt] *S5*[tg/tg] | 325 (67.6%) | 156 (32.4%) | 0 |
| ♂ *S5*[fl/fl] *S5*[tg] x ♀ *ActB-Cre S5*[del/wt] *S5*[tg] | 0 | 95 (99.0%) | 1 (1.0%) |
| ♂ *S5*[fl/fl] *S5*[tg/tg] x ♀ *ActB-Cre S5*[del/wt] *S5*[tg] | 0 | 486 (99.4%) | 3 (0.6%) |

Specifically, these are the *hCD2iCre*, *Vav1iCre* and *ActB-Cre* models carrying floxed alleles of the *Smarca5* and *S5* transgene inserted into the *Rosa26* locus.

hematopoietic stem cells and their differentiation potential in adult animals. Our model expresses *tdTomato* under the control of endogenous *Rosa26* promoter (Fig. 1a, S1e), and upon Cre recombinase activation it initiates expression of the transgenic human SMARCA5 protein with FLAG tag on its C-terminus. The reasoning for using hSMARCA5 is that it is functionally indistinguishable from murine protein. As shown in the Supplementary Fig. S1a, the human and murine SMARCA5 proteins are almost identical in amino acid sequence, and most of the differences are conservative and localized to the N-terminal unstructured region. To validate the functional proficiency of *S5*[tg], we tested its ability to rescue lymphocyte development which is blocked at progenitor stages in a conditional *hCD2iCre*-induced *Smarca5* knockout[10]. First, we confirmed that animals with the genotype *hCD2iCre S5*[tg/tg] *S5*[fl/fl] are born healthy and in expected Mendelian ratios (Table 1). Analysis of mRNA and protein levels in mouse thymus confirmed the expression of tagged SMARCA5 and revealed complete rescue at the mRNA level even by a single allele of the transgene (Fig. 1b). At the protein level a single *S5*[tg] allele provided approximately 85% of the protein compared to two endogenous *S5* alleles and two alleles of the transgene can fully rescue SMARCA5 expression (Fig. 1c). We recently reported that the complete loss of *Smarca5* gene in the *hCD2iCre* strain significantly reduces the size of the thymus and its cellularity[10]. The introduction of two hypomorphic *S5*[tg] alleles into the *hCD2iCre S5*[fl/fl] background showed an increase of thymic cellularity to almost 8-times. Despite this positive effect, the rescue was not complete and corresponded to approximately 60% of normal thymic cellularity, a similar effect was observed in splenic cellularity (Fig. 1d, S1f). Analysis of peripheral blood populations revealed almost complete lymphocyte rescue in the presence of two alleles, and one allele of the transgene

presented mild lymphopenia (Fig. 1e, S1g). To gain insight into the development of *S5* hypomorphic thymocytes, we analyzed the populations from whole thymic suspension using CD4/CD8 immunostaining by flow cytometry. Results revealed a rescue in numbers of double positive (DP) and CD4$^+$/CD8$^+$ thymocytes in *S5*[tg] animals (Fig. 1f). Upon detailed analysis of immature double negative (DN) thymocytes using CD44 and CD25 antibodies, we observed that the severe disruption of differentiation at the DN3/DN4 stage in *Smarca5* knockout (KO) was rescued in the *S5*[tg] model. This rescue was incomplete for DN1 and DN4 thymocytes, consistent with the reduced cellularity of the *S5*[tg] thymi (Fig. 1f, g).

Our previous work also revealed that SMARCA5 is required for the pro-B/pre-B transition of B cell progenitors. Indeed, the analysis of *hCD2iCre S5*[fl/fl] mice also revealed a dramatic reduction in spleen cellularity and a defect in the early B cell progenitor population, specifically a complete loss of pre-B cells (CD43$^-$CD25$^+$)[10]. The single *S5*[tg] allele did not significantly increase the number of pre-B cells, but in the presence of both *S5*[tg] alleles we observed almost complete rescue (Fig. 1h). Moreover, when we analyzed the pro-B population using CD25/CD117 markers, we observed that the *S5*[tg] was able to rescue the loss of CD25$^+$ cells but not the accumulation of CD117$^+$ cells (Fig. 1i). These data confirm that the product of the *S5*[tg] allele expressed from the ectopic locus is functional and able to almost fully rescue the lymphoid defect caused by the *hCD2iCre Smarca5* KO, which ultimately leads to normalization of peripheral blood lymphocyte numbers in animals with two copies of *S5*[tg] (Fig. 1e, S1g).

**Transgenic *Vav1-S5*[tg] cannot rescue thymocyte development in *Smarca5* null mutants**

To further study the effect of hypomorphic SMARCA5 expression on the development of thymic and splenic progenitors, we simultaneously initiated a deletion of the endogenous *Smarca5* gene and activation of *S5*[tg] in early hematopoietic precursors using *Vav1iCre*[9]. Transgenic protein level in the thymus was only ~40% of the endogenous SMARCA5 level, although their mRNA levels were comparable (Fig. 2a, b). In bone marrow, the level of transgenic SMARCA5 was even lower, at only ~18% (and ~10% from a single allele) of the level of endogenous protein (Fig. 2a, b). Importantly, in thymocytes, the expression of the hypomorphic allele on the background of the endogenous SMARCA5 protein had only an insignificant additive effect on the total protein level, suggesting the existence of a regulatory feedback loop that prevents SMARCA5 overexpression in the cells (Supplementary Fig. 2b). For these experiments, we used mice with a non-conditional variant of the SMARCA5 transgene. The transgene was activated in the parental generation by ActB-Cre recombinase. As a result, the SMARCA5 transgene is constitutively expressed in every cell in the body of the analyzed mice. Although the expression of the *S5*[tg] allele (initiated by *Vav1iCre*) rescued the embryonic lethality (Table 1), we did not observe complete establishment of definitive hematopoiesis. Specifically, transgenic animals exhibited mild anemia and, consistent with *hCD2iCre* expression model, also severe lymphopenia and reduced cellularity of thymus and spleen (Figs. 2c, d, S2c, S2d).

Lymphocyte deficiency was manifested by a decrease in the total white blood cell count (Supplementary Fig. 2d). Furthermore, *S5*[tg] expression was insufficient to promote the transition of thymocyte progenitors from DN3 to DN4 and simultaneously led to a decrease in the DP and CD4/CD8 fractions of thymocytes (Fig. 2e, f). The development of pre-B lymphocytes and pro-B cell fractions (CD117$^+$ and CD25$^+$) was similarly defective, which corresponded to a decrease in spleen cellularity (Figs. 2c, g, h, S2e). In addition, we observed the phenotypic changes that were SMARCA5 dose-dependent, as the expression of the transgene from both alleles improved the rescue of definitive hematopoiesis (Fig. 2c–h). While the changes in erythropoiesis were marginal, we found that animals expressing a single copy of the transgene showed mild anemia (lower red blood cell count, hemoglobin and hematocrit levels) and negligible changes in erythropoiesis were also observed in spleen and BM (Supplementary Fig. 2d, f–i). Overall, our data confirm the importance of SMARCA5 expression levels during hematopoiesis, with lymphopoiesis and, to some extent, erythropoiesis being dependent on high expression levels in contrast to other lineages

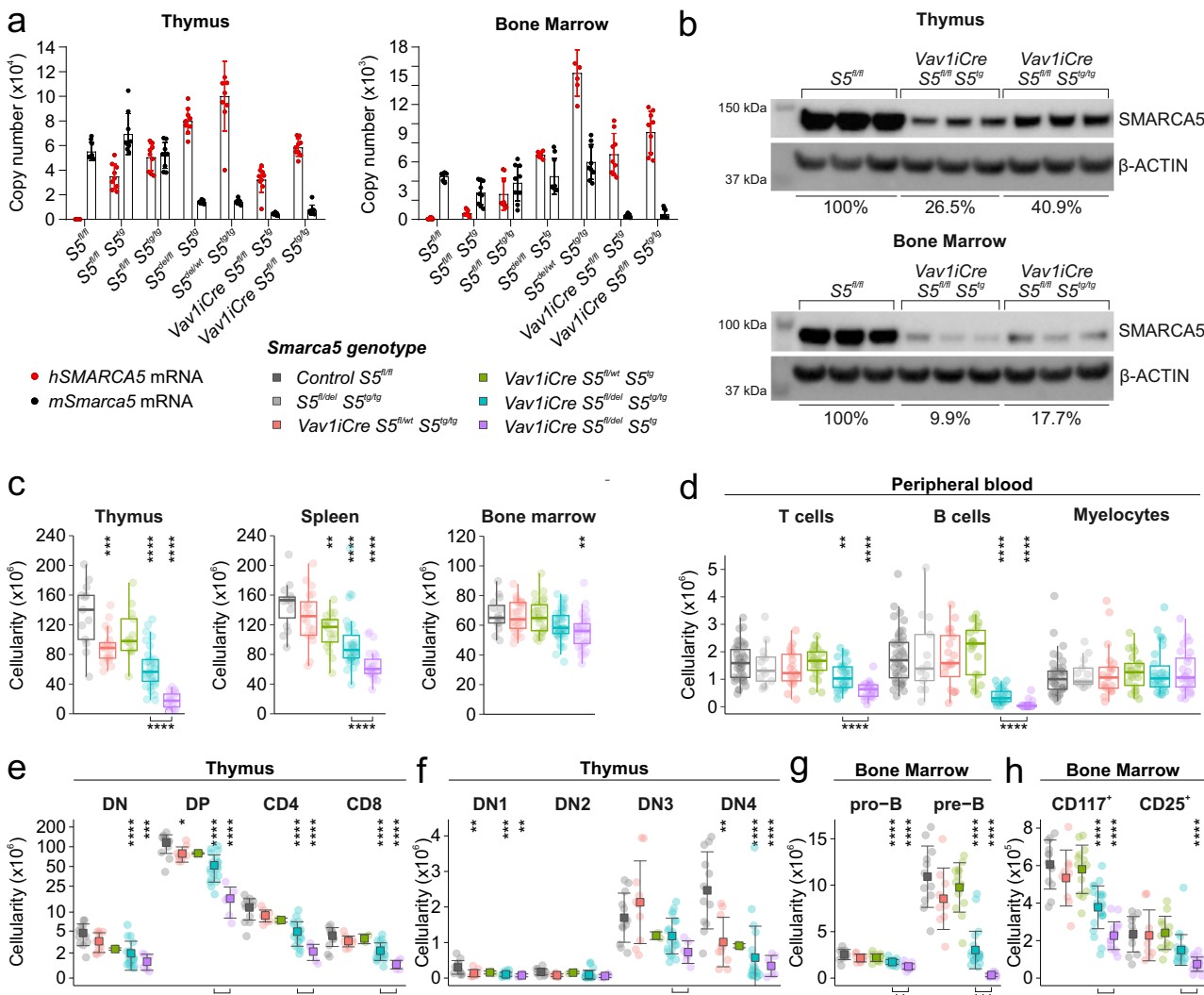

**Fig. 2 | Mice expressing *SMARCA5* transgene in stem cells with simultaneous deletion of endogenous *Smarca5* have defects in T and B cell differentiation.**
**a** Copy number quantification of mouse and human *Smarca5* mRNA by qRT-PCR (AVG ± SD, *n* = 3) in thymi and BM of 2-month-old mice of the indicated genotypes after normalization to *Gapdh* and *Hprt* expression. **b** Immunoblot of thymi and BM of 2-month-old mice of the indicated genotypes with anti-SMARCA5 antibody. Staining with β-ACTIN and GAPDH were used as loading controls. Signal density is stated in % relative to controls. **c** Thymus, spleen and BM cellularity of indicated genotypes. **d** Cytometric analysis using anti-CD4/CD8, anti-B220 staining of CD45[+] peripheral blood cells (with exclusion of Gr-1, CD11b and Nk1.1) of the indicated genotypes. **e** Cytometric analysis of DN, DP and SP populations in thymus of the

indicated genotypes. **f** Cytometric analysis of thymocytes (negative for B220, Gr-1, CD11b, CD11c and Nk1.1) of the indicated genotypes using anti-CD25 and anti-CD44 antibodies. **g, h** Cytometric analysis of B cell development of the indicated genotypes using anti-B220 and anti-CD43 staining of BM CD45[+] cells (**g**) and anti-CD117 and anti-CD25 staining of pro-B cells (CD43[+]B220[+]) (**h**). Statistics: One-Way ANOVA with Tukey's Honestly Significant Difference test, asterisks above graphs (*p* adjusted value: $p < 0.05$ = *, $p < 0.01$ = **, $p < 0.001$ = ***, $p < 0.0001$ = ****, no asterisks = non-significant). T-test between one and two alleles of $S5^{tg}$, asterisks below graphs ($p < 0.05$ = *, $p < 0.01$ = **, $p < 0.001$ = ***, $p < 0.0001$ = ****, no asterisks = non-significant), the error bars represent standard deviation, for n numbers see Supplementary Table 1.

(e.g. myelocytes – Fig. 2d), which were completely rescued even by just a single allele of $S5^{tg}$.

**Transgenic *Vav1-S5^{tg}* phenotype involves hematopoietic stem and progenitor cell compartments**
The deletion of *Smarca5* at the onset of definitive hematopoiesis (*Vav1iCre*) causes accumulation and blockade of maturation of hematopoietic stem cells (HSCs) and progenitors in the fetal liver, ultimately leading to the death of the developing embryos[9]. Since *Vav1iCre S5^{fl/fl}* individuals expressing the hypomorphic $S5^{tg}$ allele overcome this embryonic lethality, we further investigated the potential stem cell defect caused by low levels of SMARCA5 protein. First, we analyzed the composition of hematopoietic stem cells and multipotent progenitor (MPP) populations in *Vav1iCre S5^{fl/del} S5^{tg/tg}* and *Vav1iCre S5^{fl/del} S5^{tg}* animals. Bone marrow isolated from adult (8-10 weeks)

mice was analyzed by flow cytometry using stem cell markers (Sca1, c-Kit, CD34, and FLT3), and signal lymphocyte activating molecule (SLAM) antigens CD48 and CD150 (Fig. 3a, b). Animals with one copy of $S5^{tg}$ exhibited accumulation of the earliest Lin[-]Sca1[+]c-Kit[+] (LSK) population compared to $S5^{tg/tg}$ and control (Figs. 3c, S3a).

The largest defect was observed in the MPP populations of LSK cells, which consist of phenotypically distinct and long repopulating HSCs (Fig. 3d–f, S3c). The CD34[+]FLT3[-] LSK cell population was accumulated in single $S5^{tg}$ mice compared to controls, representing up to a ten-fold increase in *Vav1iCre S5^{fl/del}S5^{tg}* mice (Supplementary Fig. 3b, f). However, this net increase almost entirely consisted of MPP3 (CD48[+]CD150[-]), which can contribute to both myeloid and lymphoid lineages, and also of myeloid-committed MPP2 (CD48[+]CD150[+]) (Figs. 3e, S3c). Interestingly, lymphoid-committed MPP4 (CD48[+]CD150[-]FLT3[+]) were unaffected, but

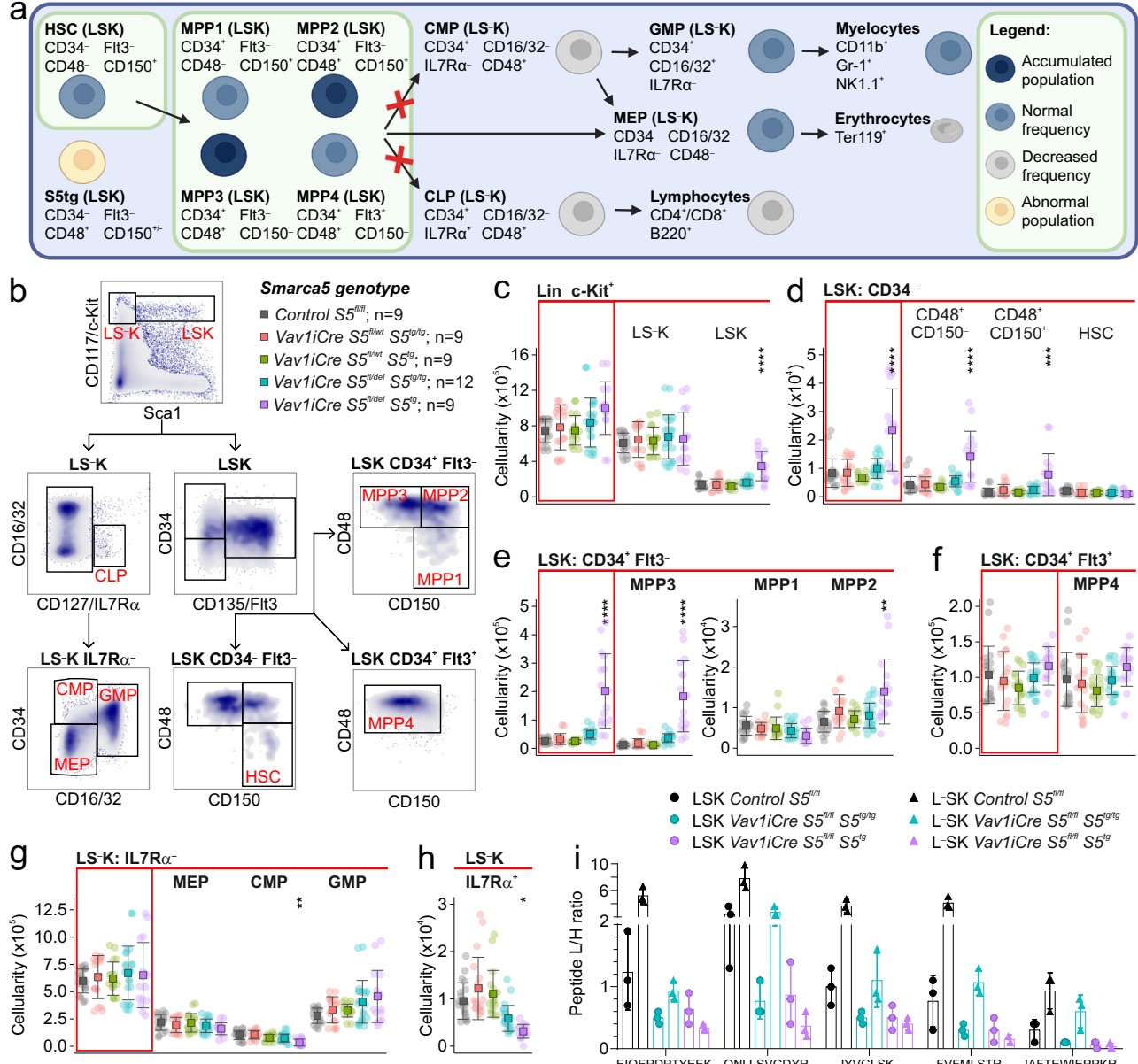

**Fig. 3 | Mice expressing *SMARCA5* transgene in stem cells with simultaneous deletion of endogenous *Smarca5* have impaired early hematopoietic development. a** Scheme of differentiation of HSC and MPP with their specific antigens in *SMARCA5* transgenic mice. Created with BioRender.com. **b** Gating strategy for flow cytometry analysis of hematopoietic progenitors in bone marrow. **c** Absolute quantification of populations from flow cytometry analysis of BM from 10-week-old mice of the indicated genotypes. Analysis of c-Kit and Sca1 markers excluding lineage antigens (CD3, Ly-6G/Ly-6C, CD11b, B220, Ter-119) yielded isolation of stem (LSK) and early progenitor (LS⁻K) cells. **d** Absolute quantification of CD34⁻ LSK, total population on the left, CD48⁺CD150⁺/⁻ and HSCs on the right. **e** Absolute quantification of CD34⁺ Flt3⁻ LSK subpopulations enriched for MPPs with multi-lineage developmental potential. **f** Absolute quantification of lymphoid-oriented

CD34⁺ Flt3⁺ MPP4 population. **g, h** Quantification of lineage-restricted (CD127/IL7Rα⁻) progenitor subpopulations (**g**) and early IL7Rα⁺ lymphoid progenitors (**h**) in LSK BM cells using myeloid marker CD16/32 and CD34. MEP megakaryocytic-erythroid progenitor, CMP common myeloid progenitor, GMP granulocyte-monocyte progenitor. **i** Bar chart depicts ratio of intensities of light over heavy peptides from protein SMARCA5 detected by targeted parallel reaction monitoring in sorted mouse LSK and LS⁻K cell populations of indicated genotypes. For n numbers and cell counts see Supplementary Table 2. Statistics: One-Way ANOVA with Tukey's Honestly Significant Difference test (p adjusted value: $p < 0.05$ = *, $p < 0.01$ = **, $p < 0.001$ = ***, $p < 0.0001$ = ****, no asterisks = non-significant), the error bars represent standard deviation.

we observed a significant reduction in the number of IL7Rα⁺ lymphoid progenitors (LP) in animals with one allele of *S5ᵗᵍ* (Figs. 3f, h, S3c, S3d). Myeloid progenitors were affected at the level of common myeloid progenitors (CMP), which were reduced in the *S5ᵗᵍ* animals, while the more differentiated MEP and GMP populations were not (Fig. 3g, S3e). We also observed some atypical populations resembling MPP2 and MPP3 but lacking CD34 expression (LSK CD34⁻FLT3⁻CD48⁺CD150⁻/CD150⁺). These populations were accumulated in single *S5ᵗᵍ* animals (Fig. 3d). The

absolute increase in stem and progenitor cells with a marked decrease in mature lymphoid progenitors and a marginal reduction in myeloid progenitors suggests that SMARCA5 depletion leads to severe impairment in the formation of differentiating cells carrying lymphoid programs.

To determine whether the described block in differentiation is related to the amount of SMARCA5 protein, we measured its levels at different stages of progenitor differentiation. The exact levels of SMARCA5 peptides were quantitated by a liquid-chromatography tandem mass spectrometry

technique using targeted parallel reaction monitoring (PRM) method. Five selected peptides of human SMARCA5 protein were reliably detected in LSK (Lin⁻Sca⁺c-Kit⁺) and LS⁻K (Lin⁻Sca1⁻c-Kit⁺) cell populations isolated by fluorescence-activated sorting from BM of mice with defined genotypes: control $S5^{fl/fl}$, $Vav1iCreS5^{fl/del}S5^{tg/tg}$, $Vav1iCreS5^{fl/del}S5^{tg}$ (Fig. 3i). In the control genotype, we observed a significant (at least twofold) increase in relative SMARCA5 levels during the transition from LSK to LS⁻K stage of differentiation. This trend was again observed in the $Vav1iCreS5^{fl/del}S5^{tg/tg}$ genotype although it showed lower relative SMARCA5 levels compared to the control. Interestingly, the $Vav1iCreS5^{fl/del}S5^{tg}$ genotype showed comparable levels of SMARCA5 in the LSK population as in the presence of both transgenic alleles, but the LS⁻K population displayed a slight decrease in the amount of the protein. This observation showed that differentiation of cells expressing a single $S5^{tg}$ allele is inefficient but still able to produce some LS⁻K cells but with lower levels of SMARCA5 (Fig. 3i).

## Transgenic $Vav1$-$S5^{tg}$ stem and progenitor cells are defective in repopulating lymphopoiesis

The accumulation of specific MPPs in $S5^{tg}$ animals may reflect their inability to enter the differentiation pathway. To test the extent to which $S5^{tg}$ stem cells are functional, we performed a competitive BM cell transplantation assay. To distinguish transgenic and wild-type hematopoietic cells, we employed a system based on the expression of different CD45 isoforms. The transgenic animals were CD45.2⁺ in contrast to CD45.1⁺ wild-type animals. First, wild-type donor BM cells were transplanted to a set of control and transgenic recipients carrying one or two $S5^{tg}$ alleles and controls. Acceptor animals were optionally irradiated (2-6 Gy). Next, we determined short- or long-term repopulating activity by a monthly assessment of lineage antigens in peripheral blood using flow cytometry. The abundance of CD45.1⁺ progeny in $S5^{tg}$ (or $S5^{tg/tg}$) mice shows that $S5^{tg}$ mice are already highly permissive even without prior irradiation and show 33% engraftment at one month and 80% engraftment four months after transplantation (Fig. 4a, S4a). Next, we analyzed the contribution of individual B/T-lymphoid and myeloid cell lineages of wild-type origin to hematopoietic recovery. Both short- and long-term (1-4 months) repopulating activity of $Vav1iCreS5^{fl/del}S5^{tg}$ recipient mice is highly permissive for T- and B-lineage repopulation, compensating for the deficiency in lymphoid cell development in the $S5^{tg/tg}$ and $S5^{tg}$ recipient mice as well as for myeloid cells in the case of $S5^{tg}$ recipients (Fig. 4b, S4b).

In the second part of the experiment, to test the repopulating activity of $S5^{tg}$ transgenic stem cells, we isolated $S5^{tg}$ BM cells and transplanted them into the control animals. $S5^{tg}$ (CD45.2⁺) BM was competitively transplanted together with wild-type (CD45.1⁺) BM cells (3:1 ratio) into lethally (8.5 Gy) irradiated wild-type recipients. Total CD45.2⁺ cell count data showed that $S5^{tg}$ HSCs with one copy of the transgene were unable to compete with wild-type HSCs, which gave rise to all repopulated hematopoietic cells (Fig. 4c). Next, we analyzed how different lineages derived from $S5^{tg}$ BM cells engraft within the normal recipient mice and observed that donor $S5^{tg}$ HSCs were unable to fully repopulate hematopoietic cell linages as their control counterparts did. While HSCs with both alleles of the $S5^{tg}$ were able to repopulate myelocytes their lymphocyte repopulation was severely affected (Figs. 4d, S4c). Consistently with our previous observations, the effect of reduced SMARCA5 protein levels was most prominent in the developing lymphocytes.

## The SMARCA5 transgenic product forms complexes and is essential for lymphopoiesis

Next, we used the ubiquitously expressed $ActB$-$Cre$ to explore whether it is possible to maintain mouse survival solely on the $S5^{tg}$ expression (in the $S5$ KO strain). To avoid problems potentially caused by transgene activation and possible delay in S5 expression, we crossed animals with already active $S5^{tg}$ (deleted $tdTomato$). Postnatally, we identified only 4 animals ($S5^{del/del}S5^{tg/tg}$) out of ~150 expected. Due to this discrepancy, we analyzed prenatal lethality of this genotype and we observed early mortality at embryonic day E13.5 and further in the fetal period at E18.5 (Table 1). However, even when

accounting for the observed prenatal lethality, we did not obtain expected Mendelian-based numbers of animals. This fact indicates a high mortality rate in the early embryonic stages.

Next, to take advantage of the fact that several $S5^{del/del}S5^{tg/tg}$ animals were born, we collected their hematopoietic tissues and analyzed the same populations as in the previous experiments. Again, we observed that the main disturbance was in the lymphoid lineage, while myelopoiesis was more or less preserved, similar to the $Vav1iCre$ model (Figs. 5a–f, S5a–m). Thus, the severe anemia in the absence of SMARCA5 that leads to disruption of early development (see previous publication[9]) de facto prevented us from finding that of all developmental pathways, it is the earliest stages of lymphopoiesis that are most sensitive to small perturbations of SMARCA5.

These findings led us to investigate functional aspects of the $Smarca5$ transgene. As noted above, we used the ubiquitously expressed $ActB$-$Cre$ to activate $S5^{tg}$ expression in the whole body on the $Smarca5^{del/wt}$ genotype background. This experimental approach allowed us to study how transgenic SMARCA5 naturally participates in its complexes in tissue-specific manner. The composition of the ISWI chromatin complex in endogenously expressed ATPase-bound regulatory subunits has been described previously[11]. Analysis of SMARCA5-copurified proteins documented that the following complexes had varying presence in specific tissues: ACF (BAZ1A) was found in all studied tissues at a high abundance, WSTF (BAZ1B) was present in fetal liver, testes, muscle, kidney, and liver at intermediate quantity, while NoRC (BAZ2A) was observed in lower abundance. Finally, SMARCA5-NURF (RBBP4) complex was detected in the brain and spleen. In addition, some ISWI complex members and known interaction partners were detected in the fetal liver (CTCF, POLE3, CHRAC1, NCL) or thymus (STAT3, STAT1, SMARCC1, RUNX1) (Figs. 5g, h, S5n, S5o). These data collectively indicate that the transgenic SMARCA5 participates in formation of all known core ISWI complexes within the transgenic mice.

## Discussion

The SMARCA5/SNF2H ATPase is one of the essential proteins involved in the regulation of DNA replication, gene expression and DNA damage response. SMARCA5 acts as a catalytic subunit of ISWI chromatin remodeling complexes, the molecular function of which depends on their individual composition. Prominent interaction partners belong to the BAZ family[3]. Our previous studies showed that mouse $Smarca5$ is an essential gene for early myelopoiesis and erythropoiesis[9], and lymphopoiesis[10], an observation consistent with the high expression of $Smarca5$ during these stages. In this report, we present a new mouse model of transgenic $SMARCA5$ ($S5^{tg}$) expression in the context of the $Smarca5$ knockout (KO) strain. $S5^{tg}$ not only rescued embryonic lethality of $S5$ KO, but also allowed us to study the effect of its graded expression. Overall, these results provided a solid basis for understanding the requirements for $Smarca5$ gene expression levels during hematopoiesis.

Knockout in the lymphoid lineage ($hCD2iCre$) revealed that SMARCA5 is necessary for the transition between DN3 and DN4 stages of thymocyte development. At this stage, immature T lymphocytes undergo β-selection, which is accompanied by changes in gene expression. In the $hCD2iCre$ $S5$ knockout thymus, the DN3 population accumulates due to a developmental block that is followed by apoptosis. Similarly, B cell development is arrested during the transition from the pro- to the pre-B stage[10]. Our hypomorphic transgene rescued both of these situations to some extent, demonstrating that the essential role of SMARCA5 in later lymphoid progenitors depends on its presence but does not require its high expression. Neomorphic alleles of $Pole3$, a SMARCA5 interactor in CHRAC ISWI complex, revealed similar phenotypes showing arrest between DN3 and DN4 stages as well as during B cell differentiation, suggesting that the CHRAC complex is the most important for lymphoid differentiation[12]. However, knockouts of other SMARCA5-binding partners did not exhibit any defects associated with lymphopoiesis. In detail, $Acf1$ ($Baz1a$) is not essential for hematopoiesis[13]. $Tip5$ ($Baz2a$), $Chrac1$ and $Baz2b$ KO showed no significant defect in lymphopoiesis, although some of these studies did

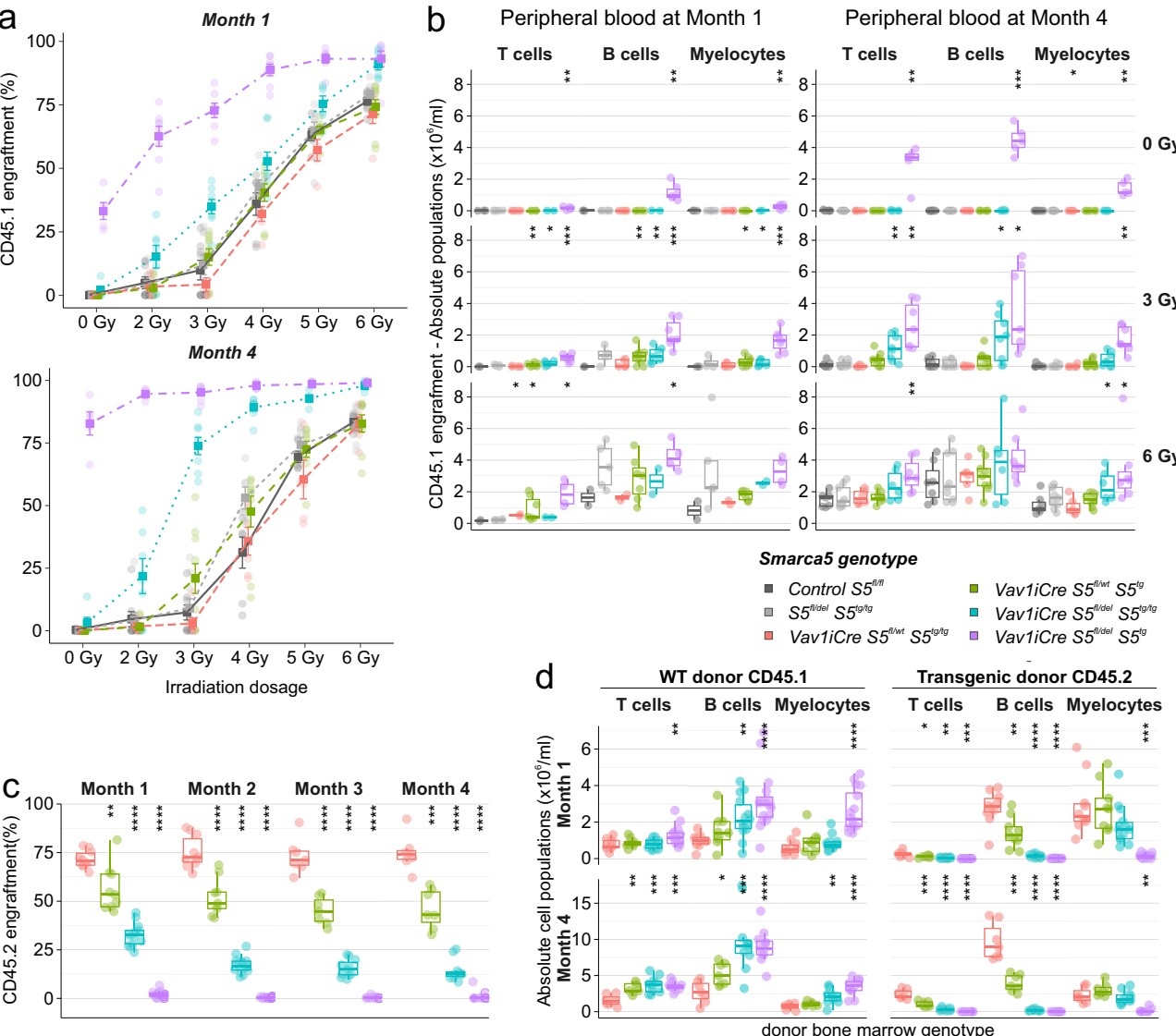

**Fig. 4 | Transplantation of stem cells with only transgenic *SMARCA5* expression reveals defects in T and B cell repopulation. a** Dose dependence of SMARCA5 expression levels of the indicated genotypes on graft repopulation (-axis). Cytometry (%) of reconstitution of control CD45.1 donor BM cells in $S5^{tg}$ recipient mice on CD45.2 background at 1 (short-term repopulation) and 4 months (long-term) after transplantation depending on irradiation dosage. **b** Absolute quantification (Y-axis) of engraftment of wild-type hematopoietic cells in BM niches of the indicated genotypes. Quantification of hematopoietic lineage progeny (T, B, and myeloid cells, X-axis) after reconstitution of CD45.1 BM donor cells from peripheral blood by flow cytometry at 1 and 4 months after transplantation depending on irradiation dosage.

**c** Relative quantification of competitive transplantation (3:1 ratio) of BM from *SMARCA5* transgenic mice (CD45.2, indicated genotypes) along with wild-type CD45.1 into lethally (8.5 Gy) irradiated wild-type CD45.1 recipients at 1-4 months. **d** Absolute quantification of competitive hematopoietic BM engraftment of the indicated genotypes. Peripheral blood progeny (T, B, and myeloid cells, X-axis) after reconstitution at 1 and 4 months (see (**c**) for details). Statistics: T-test relative to controls ($p < 0.05 = *$, $p < 0.01 = **$, $p < 0.001 = ***$, $p < 0.0001 = ****$, no asterisks = non-significant), the error bars represent standard deviation, for n numbers see Supplementary Table 3.

not include a thorough assessment of hematopoiesis[14]. *Rsf1* and *Bptf* KO are embryonic lethal[14,15]. *Cecr2* KO exhibits a variety of defects (coloboma, microphthalmia, and skeletal, heart, and kidney defects), but none of them were described in hematopoiesis[16]. Interestingly, we detected POLE3 in the SMARCA5 interactome in fetal liver, but we did not detect RSF1.

To better understand how *Smarca5* deletion affects early hematopoiesis, we used *Vav1iCre S5* KO mouse model. As shown previously, this model exhibits embryonic lethality at E18, which correlates with severe anemia and accumulation of hematopoietic progenitors[9]. We observed that the $S5^{tg}$ rescues this lethality, but transgenic animals exhibit severe lymphopenia and mild anemia. The severity of these defects was inversely related to the copy number of the $S5^{tg}$, indicating a dependence on higher SMARCA5 expression in hematopoiesis. The only population that was fully

rescued by the $S5^{tg}$ were the myelocytes. The most severe phenotype was associated with an arrest during differentiation of MPP3 to MPP4. This transition is associated with FLT3 activation which is required for lymphopoiesis. $S5^{tg}$ can rescue accumulation of MPPs if expressed from both alleles. Expression from a single allele resulted in the accumulation of cells in the MPP3 stage, and to some extent, the MPP2 stage as well. Interestingly, we also observed the accumulation of populations in transgenic animals which have a unique mixture of progenitor markers. Although they resemble the MPP2/3 populations, they are noticeably lacking the CD34 antigen. We hypothesize that these populations likely originate from the accumulated MPP2/3 which are unable to differentiate. Moreover, in this genotype, defects in lymphocyte development continued to the later stages of differentiation (DN3 to DN4 and pro-B to pre-B block), resulting in

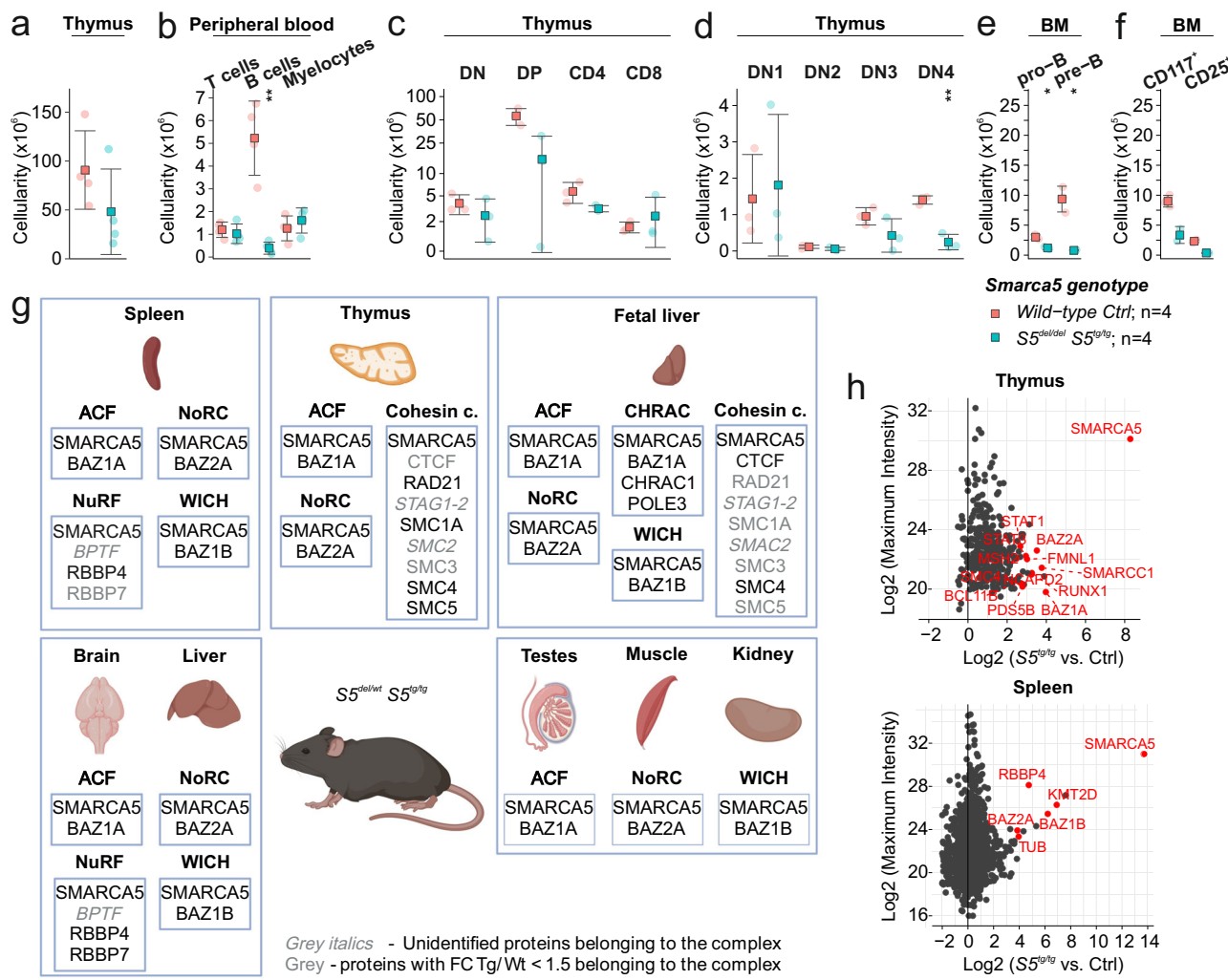

**Fig. 5 | The SMARCA5 transgenic product forms complexes and is essential for lymphopoiesis. a** Thymus cellularity in 1-year-old mice of the indicated genotypes (analyzed by cytometry). **b** Flow cytometry analysis with anti-CD4/CD8, anti-B220, and anti-Gr-1/CD11b antibodies from CD45$^+$ peripheral blood cells of the indicated genotypes. **c** Cytometric analysis of thymic populations: DN, DP, and SP of the indicated genotypes. Gr-1$^+$, CD11b$^+$, and Nk1.1$^+$ cells were excluded. **d** Cytometric analysis of thymocytes (negative for B220, Gr-1, CD11b, CD11c and Nk1.1) of the indicated genotypes using anti-CD25 and anti-CD44 antibodies. **e, f** Cytometric analysis of B cell development of the indicated genotypes using anti-B220 and anti-CD43 staining of BM CD45$^+$ cells (**e**) and anti-CD117 and anti-CD25 staining of

pro-B cells (CD43$^+$B220$^+$) (**f**). Statistics: t test relative to controls ($p < 0.05 = *$, $p < 0.01 = **$, $p < 0.001 = ***$, $p < 0.0001 = ****$, no asterisks = non-significant), the error bars represent standard deviation. **g** SMARCA5-copurified proteins identified by mass spectrometry in mouse hematopoietic (spleen, thymus, fetal liver) and other organs and their involvement in complexes with SMARCA5 (ACF, NoRC, NuRF, WICH, CHRAC and Cohesin complex). Created with BioRender.com. **h** Mass spectrometric analysis of SMARCA5 complexes in represented organs. Tissues from 2-month-old wild-type (negative control, $n = 3$) and transgenic ($n = 3$) animals were lysed and tagged SMARCA5 was immunoprecipitated using FLAG-M2 agarose beads.

almost complete loss of lymphocytes in peripheral blood, with B cells being more affected than T cells. These results were consistent with our observations from the *hCD2iCre S5*$^{tg}$ model, but the defect in *Vav1iCre S5*$^{tg}$ model was more pronounced, suggesting that this is a stem cell-level phenotype. The observations from flow cytometry were also supported by measurements of SMARCA5 protein levels in different developmental stages of hematopoietic progenitors. We observed that transition from LSK to the later developmental stage LS$^-$K was accompanied by an enhancement of SMARCA5 protein levels in control mice, whereas in the presence of only one allele of the transgene, there was a slight loss of protein. The reduced levels of SMARCA5 likely result in a decrease in CMP and CLP populations (LS$^-$K) in this genotype and an accumulation of MPP3 and MMP2 populations (LSK). These observations imply that *Smarca5* expression undergoes independent regulation from its promoter during hematopoietic progenitor differentiation. Prior research has demonstrated that SMARCA5 can be subjected to degradation through Cullin- and proteasome-dependent pathways[17]. Although this represents an interesting and novel concept that is

clearly important for hematopoietic differentiation, we currently lack data that unequivocally support it. Interestingly, we observed a discrepancy between SMARCA5 transgene expression at the RNA and protein levels, with the transgene being able to reach endogenous *Smarca5* RNA level but not the protein level suggesting a regulatory mechanism for SMARCA5 at the protein level requiring further investigation.

To support our hypothesis that SMARCA5 expression level has a direct effect on HSCs, we employed competitive transplantation of wild-type HSC into *S5* transgenic acceptors and vice versa. Both of these experiments confirmed our hypothesis that high levels of SMARCA5 are important for lymphocyte development, as wild-type HSCs reconstituted lymphopoiesis in animals with a single *S5*$^{tg}$ allele even without pre-transplantation irradiation. *S5*$^{tg}$ BM was only able to reconstitute myelocytes but not lymphocytes in lethally irradiated wild-type acceptors. The effects observed in competitive transplantation were clearly dose-dependent, with the most substantial impact seen when a single *S5*$^{tg}$ allele was expressed. A similar competitive transplantation experiment performed with neomorphic *Pole3*

(member of ISWI CHRAC complex) and wild-type BM (1:1) revealed that HSCs with a mutated *Pole3* struggled to repopulate the lymphoid lineage but not the myeloid lineage[12]. Deletion of ARID1A in mouse (SWI/SNF complex) resulted in a similar but more severe phenotype with an accumulation of hematopoietic progenitors resulting in defects in lymphopoiesis, myelopoiesis and erythropoiesis. These accumulated progenitors were not able to repopulate host hematopoiesis in a competitive transplantation experiment[18].

The final experiment focused on using a whole-body deletion of *Smarca5* combined with *S5^{tg/tg}* expression to determine whether any other processes during development require higher than transgenic SMARCA5 protein levels. We observed a similar phenotype as in the *Vav1iCre* model with a defect in hematopoiesis combined with very low birth expectancy – less than 1% (contrasted with the expected value of 25%). Most of these animals died in the early embryonic stages, suggesting their lower fitness.

As in our experiments, knockouts of SWI/SNF chromatin remodeling factors were found to cause defects in hematopoiesis. For example, in a mouse model with mutant *Baf57* (disruption of DNA binding), CD8$^+$ T cell development was blocked even though only one allele was mutated. The same block in differentiation was observed in mice with a heterozygous deletion of *Brg1*[19]. In contrast to our findings from *Smarca5* KO, *Brg1* loss results in a block of T cell differentiation at later stages, the DN4 to DP transition[20]. Another chromatin remodeling factor with a significant role in hematopoiesis is Mi-2β (Mi-2 family) which is essential for the DN4 to DP transition and especially for CD4$^+$ T cell development[21].

The model of whole-body SMARCA5 expression allowed us to study the composition of its ISWI complexes in a tissue-specific manner. While our results are dominated by canonical complexes with BAZ proteins (as updated in[11]), it certainly cannot be ruled out that there may be other interactions with DNA at the chromatin level that are not easily detected by the mass spectrometry methodology we used. These non-canonical complexes might be context-dependent and difficult to detect in a mixture of cell populations. Nevertheless, our data also show the involvement of CTCF in the ISWI complex in some tissues, particularly in fetal hematopoiesis as suggested elsewhere[6].

## Materials & methods
### Mouse transgenesis
*SMARCA5* cDNA fused to a C-terminal *FLAG Tag* (cleaved from MSCV2.2 vector) was cloned into *Nhe*I and *Sal*I cleavage sites of the p15-6-20 DNA vector containing 5' & 3' homologies of *Rosa26* and the *tdTomato* gene surrounded by LoxP sites, allowing excision of *tdTomato* and initiation of *SMARCA5* transgene expression (Fig. 1a)[22]. The assembled transgenic vector (linearized by *Fse*I) was nucleofected (Amaxa Nucleofector II instrument, program A-023, Lonza) into R1 ES cells ($2 \times 10^6$) which were seeded on a layer of irradiated (35 Gy) mouse embryonic fibroblasts in recommended media (details below) supplemented with ESGRO (mLIF 1 kU/ml, cat. no. ESG1107, Merck, Darmstadt, Germany)[23]. G418-resistant colonies (200 µg/ml, cat. no. A1720, Sigma-Aldrich, Saint Louis, MO, USA) expressing *tdTomato* were selected and expanded. A 3.495 kbp PCR band (F: GTGTCTCTTTTCTGTTGGACCCTTACCT; R: AAAAAGAA-GAAGGCATGAACATGGTTAG) verified integration of the transgene (Supplementary Fig. 1b). ES cells with a normal karyotype were injected into blastocysts and transferred into the uterus of superovulated *C57BL/6NCrl* females. Mice containing the *Rosa26-S5^{tg}* construct were identified using PCR with the following primers Fwd: ACGAGGACCAAAGCCTTCAA-CAG, Rev: TGACAGTCGACAACTTCCTGACTAGGGAGTAGAAGT (Supplementary Fig. 1c). Germline transfer and backcrossing was performed into the C57BL/6NCrl strain. The progeny was viable, fertile and expressed *tdTomato* in all tissues. Mice expressing *tdTomato* with Cre-inducible SMARCA5 transgene were then crossed to three Cre expressing strains (*hCD2* which is active from DN3 stage of thymocyte development and pro-B stage of B-cell development and CD2$^+$ NK cells[10,24], *Vav1* promoting recombination in c-kit$^+$(CD117$^+$) definitive

hematopoietic progenitors[9], and constitutively expressed *Gt(ROSA) 26Sor^{tm1(ACTB-cre,-EGFP)Ics}*[25]) containing a conditional knockout allele of *Smarca5* with *loxP1* flanking (fl) exon 5 of the endogenous *Smarca5* gene, resulting in a null allele upon Cre activation[9,10].

All mice expressing Cre recombinase were hemizygous for specific Cre transgene. Mice were maintained in a controlled, specific pathogen-free environment. Mice were provided food and water *ad libitum* and kept in the animal facility with a 12-hour light-dark cycle. All experiments met the criteria approved by Czech Ministry of Agriculture and the Committee for Experimental Animals. We have complied with all relevant ethical regulations for animal use. In all experiments, we used mice of both sexes, age is indicated in figure legends.

### ES cells culture medium
KnockOut™ DMEM (cat. no. 10829018, Gibco, Thermo Fisher Scientific, Waltham, Massachusetts, United States) supplemented with 15% Knock-Out™ Serum Replacement (cat. no. 10828028, Gibco, Thermo Fisher Scientific, Waltham, Massachusetts, United States), 2 mM L-glutamine (cat. no. G7513, Sigma-Aldrich, St. Luis, MO, USA), 1x Penicillin-Streptomycin (cat. no. P0781, Sigma-Aldrich, St. Luis, MO, USA), 1x MEM Non-Essential Amino Acid Solution (cat. no. M7145, Sigma-Aldrich, St. Luis, MO, USA), 0,1 mM β-mercaptoethanol (cat. no. M3148, Sigma-Aldrich, St. Luis, MO, USA), ESGRO mLIF (1 kU/ml, cat. no. ESG1107, Merck, Darmstadt, Germany).

### Tail tip DNA was genotyped with the following primers
S5^{tg}-F: ACGAGGACCAAAGCCTTCAACACAG, R: TGACAGTCGA-CAACTTCCTGACTAGGGGAGGAGTAGAAGT (852 bp), S5^{fl}-F: ACT-GAGGACTCTGATGCAAACAGTCAAG, R: TACACAACTAAGGCAG TGGGTTATAGTGC (fl 614 bp, wt 524 bp), S5^{del}-F: GTGCAAAGCCCAG AGACGATGGTATG (471 bp, with S5^{fl}-rev), Cre-F: ACCAGGTTCGTT-CACTCATGG, Cre-R: ACGGGCACTGTGTCCAGACC (449 bp), iCre-F: GATGCTCCTGTCTGTGTGCAG, iCre-R: CCTGCCAATGTGGAT-CAGC (469 bp).

### Cell isolation and analysis
Single cell suspensions from peripheral blood, BM (isolated from 2 femurs and 2 tibias), thymus, and spleen were incubated on ice with specific primary antibodies in PBS + 1% biotin-free BSA solution for 20 min. Biotinylated primary antibodies were then visualized using streptavidin-conjugated fluorescent dyes. Cell suspensions were analyzed on BD LSRFortessa™ SORP (BD Biosciences, San Jose, CA, USA) or CytoFLEX (Beckman Coulter, Brea, CA, USA), and data analysis was performed using FlowJo software.

### Antibodies for flow cytometry
Biotin anti-B220 (clone RA3-6B2; cat. no. 103204), FITC anti-B220 (clone RA3-6B2; cat. no. 103206), APC anti-CD4 (clone GK1.5; cat. no. 100412), BV421 anti-CD4 (clone GK1.5; cat. no. 100437), PE/Cy7 anti-CD5 (clone 53-7.3; cat. no. 100621), APC anti-CD8a (clone 53-6.7; cat. no. 100712), Biotin anti-CD11b/Mac-1 (clone M1/70; cat. no. 101204), BV421 anti-CD11b/Mac-1 (clone M1/70; cat. no. 101236), Biotin anti-CD11c (clone N418; cat. no. 117304), BV510 anti-CD16/32 (clone 93; cat. no. 101333), BV605 anti-CD25 (clone PC61; cat. no. 102036), PE anti-CD25 (clone PC61; cat. no. 102008), Biotin anti-CD34 (clone HM34; cat. no. 128604), APC anti-CD43 (clone S11; cat. no. 143208), FITC anti-CD44 (clone IM7; cat. no. 103006), PE/Cy7 anti-CD45 (clone 30-F11; cat. no. 103114), AF700 anti-CD45.1 (clone A20; cat. no. 110724), PE/Cy7 anti-CD45.2 (clone 104; cat. no. 109830), FITC anti-CD48 (clone HM48-1; cat. no. 103404), PE anti-CD71 (clone RI7217; cat. no. 113808), BV421 anti-CD117/c-Kit (clone 2B8; cat. no. 105828), BV786 anti-CD127/IL7Ra (clone A7R34; cat. no. 135037), APC anti-CD135/FLT3 (clone A2F10; cat. no. 135310), BV605 anti-CD150 (clone TC15-12F12.2; cat. no. 115927), Biotin anti-Gr-1 (clone RB6-8C5; cat. no. 108404), BV605 anti-Gr-1 (clone RB6-8C5; cat. no. 108440), Biotin anti-Nk1.1 (clone PK136; cat. no. 108704), PE anti-Nk1.1 (clone PK136; cat.

no. 108708), FITC anti-Sca1 (clone D7; cat. no. 108106), PE anti-Sca1 (clone D7; cat. no. 108108), APC anti-Ter119 (clone Ter-119; cat. no. 116212), Biotin anti-Ter119 (clone Ter-119; cat. no. 116204), PerCP anti-Ter119 (clone Ter-119; cat. no. 116226), AF700 anti-mouse Lineage Cocktail with Isotype Ctrl (anti-CD3, clone 17A2; anti-Ly-6G/Ly-6C, clone RB6-8C5; anti-CD11b, clone M1/70; anti-CD45R/B220, clone RA3-6B2; anti-TER-119/Erythroid cells, clone Ter-119; cat. no. 133313), SV-PE/Cy7 (cat. no. 405206), SV-AF700 (cat. no. S21383; Life Technologies Corporation, Eugene, Oregon, USA). All antibodies supplied by BioLegend (San Diego, California, USA) with exception for SV-AF700.

## Immunoblotting

Organs from 7–9-week-old mice were homogenized and lysed on ice for 30 min in isotonic lysis buffer (150 mM NaCl, 50 mM Tris-Cl pH 7.5, 0.4% Triton-X, 2 mM $CaCl_2$, 2 mM MgCl2, 1 mM EDTA, 5 mM NaF in $dH_2O$), supplemented with 1 mM DTT, protease and phosphatase inhibitors and 25 U/μl of non-specific DNA nuclease (Benzonase; cat. no. SC-391121, Santa Cruz, CA, USA). Lysates were then mixed with SDS (final 1%) was heated for 5 min, 95 °C. Lysates were cleared by centrifugation (16 000 g, 4°C, 5 min) and protein concentration was determined by bicinchoninic acid assay (cat. No. 23228, Thermo Fisher, Waltham, MA, USA). 20 μg of protein was separated on a 4-20% SDS gradient of Mini-PROTEAN TGX Precast Protein gels and transferred with the Trans-Blot Turbo Transfer System to a PVDF membrane (all from Bio-Rad, Hercules, CA, USA). Membranes were blocked in 5% milk in TBS-T and incubated (overnight, 4 °C) in primary antibodies diluted in 3% BSA in TBS-T. After washing, membranes were incubated with horseradish peroxidase-conjugated anti-Rabbit IgG secondary antibody (1:10,000; Jackson ImmunoResearch, Cambridgeshire, UK; cat. no. 711-036-152) or with anti-Mouse IgG secondary antibody (1:10,000; Jackson ImmunoResearch, Cambridgeshire, UK; cat. no. 715-036-150). Primary antibodies: anti-SMARCA5 (1:1,000; Bethyl Laboratories, Montgomery, TX, USA; cat. no. A301-017A), anti-FLAG (1:1,000; Cell Signaling, Danvers, MA, USA; cat. no. 2368), anti-β-ACTIN (1:1,000; Abcam, Cambridgeshire, UK; cat. no. ab6276) and anti-GAPDH (1:1,000; Sigma-Aldrich, Saint Louis, MO, USA; cat. no. HPA040067). The binding efficiency of anti-SMARCA5 antibody with human and murine protein was tested using leukemia cell lines, specifically MEL (murine erythroleukemia) OCI-M2 (adult human acute myeloid leukemia) (Supplementary Fig. 1d). Visualization was performed using Pierce™ ECL Western Blotting Substrate (cat. no. PI32106, Thermo Fisher, Waltham, MA, USA) and detection using ChemiDoc Imaging System (Bio-Rad, Hercules, CA, USA). For densitometry all samples were normalized to their β-ACTIN/GAPDH loading control and the relative amount of protein was calculated compared to wild-type controls. 5. Uncropped and unedited blot images are available in Supplementary Fig. 6.

## Hematopoietic reconstitution

For competitive reconstitution experiments $7.5 \times 10^5$ BM cells from wild-type adult (12-week-old) *B6 SJL-Ptprc^a Pep^b/BoyCrl* (*Ly5.1*/CD45.1) donors were co-transplanted with $2.25 \times 10^6$ BM cells (ratio 1:3, wt:tg) from *C57Bl/6* (*Ly5.2*/CD45.2) *SMARCA5* transgenic donors and their respective controls into lethally irradiated (8.5 Gy) adult (12-week-old) *Ly5.1* mice. In addition, hematopoiesis was assessed in *SMARCA5* transgenic hosts by transplanting $3 \times 10^6$ BM cells collected from wild-type 12-week-old *Ly5.1* mice into non-irradiated or irradiated (2, 3, 4, 5, up to 6 Gy) *Ly5.2 SMARCA5* transgenic hosts and controls. Hematopoietic repopulation was analyzed using flow cytometry analysis of peripheral blood collected at monthly intervals (1 to 4 months) after transplantation.

## RT-PCR

RNA from homogenized organs was purified with TRIzol™ (cat. no. 15596026, Thermo Fisher, Waltham, MA, USA), treated with DNA-free™ DNA Removal Kit (cat. no. AM1906, Thermo Fisher, Waltham, MA, USA) and reverse transcribed with High-Capacity cDNA Reverse Transcription Kit (cat. no. 4374966, Thermo Fisher, Waltham, MA, USA). qPCR

amplification using LightCycler® 480 SYBR Green I Master (cat. no. 04887352001, Roche, Basel, CH). Primers: *mS5* F: AGAATTTGCTTT-CAGTTGGAGATTACCG, *mS5* R: AGATGAGCCAATTCAATCCTCG C, *hS5* F: AGAACTTACTATCCGTTGGCGATTACC; *hS5* R: AAGAAA TGAGCCAGTTTAATCCTCGG, *mGapdh* F: ACTTTGTCAAGCTCAT TTCCTGGTATG-3′, *mGapdh* R: TTTCTTACTCCTTGGAGGCCATG TAG, *mHprt* F: GCTGGTGAAAAGGACCTCT, *mHprt* R: CACAGGAC TAGAACACCTGC.

## Protein immunoprecipitation

Organs from 7–9-week-old mice ($S5^{del/wt} S5^{tg/tg}$ and controls) were rapidly frozen using liquid nitrogen, homogenized and lysed in an isotonic lysis buffer (150 mM NaCl, 50 mM Tris-HCl pH=7.5, 0.4% Triton X-100, 2 mM $CaCl_2$, 2 mM $MgCl_2$, 1 mM EDTA, 5 mM NaF in $ddH_2O$) with Benzonase nuclease (250 U/ml, cat.no. SC-391121B, Santa Cruz Biotechnology, Texas, USA), protease and phosphatase inhibitors and 10 mM $Na_3VO_4$ for 20 min on ice. After sonication, tissue lysates were further cleared by ultra-centrifugation (20,000 g, 4 °C, 10 min) and filtration using a 0.45 μm filter. Individual tissue protein concentrations were equilibrated and lysates corresponding to 100-250 mg of tissue were incubated with 50 μl of anti-FLAG M2 Affinity Gel (cat. no. A2220, Sigma-Aldrich, St. Luis, MO, USA) for three hours. The immunoprecipitated complexes were washed four times with a lysis buffer and eluted with 3xFLAG peptide (cat. no. F4799, Sigma-Aldrich, St. Luis, MO, USA) in a detergent-free lysis buffer. The eluates were then analyzed by mass spectrometry.

## Mass spectrometry details

Eluates were acetone precipitated and resuspended in 100 mM TEAB containing 1% SDC. Cysteines were reduced with 5 mM final concentration of TCEP (60°C for 60 min) and blocked with 10 mM final concentration of MMTS (10 min, RT). Samples were cleaved on beads with 1 μg of trypsin at 37 °C overnight. After digestion samples were centrifuged and supernatants were collected and acidified with TFA to 1% final concentration. SDC was removed by extraction to ethyl acetate[26]. Peptides were desalted using in-house made stage tips packed with C18 disks (Empore) according to[27].

Nano Reversed phase column (EASY-Spray column, 50 cm × 75 μm ID, PepMap C18, 2 μm particles, 100 Å pore size) was used for LC/MS analysis. Mobile phase buffer A was composed of water and 0.1% formic acid. Mobile phase B was composed of acetonitrile and 0.1% formic acid. Samples were loaded onto the trap column (Acclaim PepMap300, C18, 5 μm, 300 Å Wide Pore, 300 μm × 5 mm, 5 Cartridges) for 4 min at 15 μl/min. Loading buffer was composed of water, 2% acetonitrile and 0.1% trifluoroacetic acid. Peptides were eluted with Mobile phase B gradient from 4% to 35% B in 60 min. Eluting peptide cations were converted to gas-phase ions by electrospray ionization and analyzed on a Thermo Orbitrap Fusion (Q-OT-qIT, Thermo). Survey scans of peptide precursors from 400 to 1600 m/z were performed at 120 K resolution (at 200 m/z) with a $5 \times 10^5$ ion count target. Tandem MS was performed by isolation at 1,5 Th with the quadrupole, HCD fragmentation with normalized collision energy of 30, and rapid scan MS analysis in the ion trap. The $MS^2$ ion count target was set to $10^4$ and the max injection time was 35 ms. Only those precursors with charge state 2–6 were sampled for $MS^2$. The dynamic exclusion duration was set to 45 s with a 10 ppm tolerance around the selected precursor and its isotopes. Monoisotopic precursor selection was turned on. The instrument was run in top speed mode with 2 s cycles[28].

All data were analyzed and quantified with the MaxQuant software (version 1.5.3.8)[29]. The false discovery rate (FDR) was set to 1% for both proteins and peptides and we specified a minimum length of seven amino acids. The Andromeda search engine was used for the MS/MS spectra search against the Caenorhabditis elegans database (downloaded from Uniprot in April 2015, containing 25 527 entries). Enzyme specificity was set as C-terminal to Arg and Lys, also allowing cleavage at proline bonds and a maximum of two missed cleavages. Dithiomethylation of cysteine was selected as fixed modification and N-terminal protein acetylation, methionine oxidation and serine/threonine/tyrosine phosphorylation as variable

## Table 2 | List of peptides used for PRM detection of protein SMARCA5

| Compound | m/z | z |
|---|---|---|
| EIQEPDPTYEEK | 739.3383 | 2 |
| EIQEPDPTYEE**K** | 743.3454 | 2 |
| QNLLSVGDYR | 582.804 | 2 |
| QNLLSVGDY**R** | 587.8082 | 2 |
| IYVGLSK | 390.2367 | 2 |
| IYVGLS**K** | 394.2438 | 2 |
| FVFMLSTR | 500.7679 | 2 |
| FVFMLST**R** | 505.772 | 2 |
| IAFTEWIEPPKR | 496.2733 | 3 |
| IAFTEWIEPPK**R** | 499.6094 | 3 |

Peptides precursor mases (m/z) and charge states of the precursor ions (z) are indicated. Heavy amino acids are labeled by bold letter.

modifications. The "match between runs" feature of MaxQuant was used to transfer identifications to other LC-MS/MS runs based on their masses and retention time (maximum deviation 0.7 min) and this was also used in quantification experiments. Quantifications were performed with the label-free algorithms described recently. Data analysis was performed using Perseus 1.5.2.4 software.

### Targeted parallel reaction monitoring of SMARCA5 peptides

Sorted cells from LSK (Lin⁻c-Kit⁺Sca⁺) and LS⁻K (Lin⁻c-Kit⁺Sca1⁻) cell populations were washed with PBS and cell pellet was lysed with 0.1% RapiGest (Waters) in 50 mM ammonium bicarbonate (Sigma Aldrich). Protein cysteines were reduced with 5 mM dithiothreitol (Thermo Scientific) for 30 min at 56 °C and alkylated with 10 mM iodoacetamide (Sigma Aldrich) for 30 min at room temperature in the dark. Proteins were digested with sequencing grade modified trypsin (Promega) (enzyme:protein ratio 1:50) at 37 °C over night. The reaction was stopped by adding trifluoroacetic acid to a final concentration of 1% (v/v), and RapiGest was precipitated by further incubation at 37 °C for 45 min. Supernatant was cleaned up using solid phase extraction stage tips and dried by vacuum centrifugation. Peptide digests were spiked in with equivalent amount of heavy peptides premix normalized to the used cell count (Supplementary Table 2).

Five peptides used for targeted parallel reaction monitoring (PRM) using mass spectrometry were selected according to reliable detection in shotgun analysis fulfilling these criteria (1) unique; (2) no missed cleavages (3) not containing cysteine; (4) no modifications and using Picky[30] (Table 2). Heavy peptides were purchased from JPT Peptide (JPT Technologies, Berlin, Germany). Cell lysates with spiked in heavy standards were first separated using Ultimate 3000 liquid chromatography system (Dionex). Nano reversed phase columns (Aurora Ultimate TS, 25 cm × 75 μm ID, 1.7 μm particle size, Ion Opticks) were used for LC/MS analysis. Mobile phase buffer A was composed of water and 0.1% formic acid. Mobile phase B was composed of acetonitrile and 0.1% formic acid. Samples were loaded onto the trap column (C18 PepMap100, 5 μm particle size, 300 μm × 5 mm, Thermo Scientific) for 4 min at 18 μl/min loading buffer was composed of water, 2% acetonitrile and 0.1% trifluoroacetic acid. Peptides were eluted with Mobile phase B gradient from 4% to 35% B in 16 min. Total analysis time was 30 min per sample. Samples were analyzed by PRM on Orbitrap Ascend (Thermo Scientific). Eluting peptide cations were converted to gasphase ions by electrospray ionization in positive mode with 1600 V spray voltage. MS data were acquired using tMS2 mode. RF lens amplitude was set to 60%. Peptide precursors were isolated by quadrupole with 1.6 m/z isolation window and fragmented by HCD with collision energy set to 28%. Fragment ions were detected in orbitrap with 30 K resolution at 200 m/z. AGC was set to 250% and maximum injection time mode to Auto. Total

cycle time was set to 0.8 s. All peptides with their corresponding masses and charge states are listed in the Table 2. Data were analyzed using Skyline-daily (64-bit) version 23.1.1.268. Identical heavy and light transitions and retention times confirmed peptide identity. A minimum of four transitions was required for reliable detection. All peaks were manually inspected to confirm correct detection and peak boundaries. Peak integration and calculation of the ratios between light endogenous and the heavy-labeled peptide (L/H) were done in Skyline and result reports exported from the software.

### Sequence coverage by selected peptides

>sp|O60264|SMCA5_HUMAN SWI/SNF-related matrix-associated actin-dependent regulator of chromatin subfamily A member 5 OS=Homo sapiens OX = 9606 GN = SMARCA5 PE = 1 SV = 1.

MSSAAEPPPPPPPESAPSKPAASIASGGSNSSNKGGPEGVAAQA-VASAASAGPADAEMEEIFDDASPGKQK**EIQEPDPTYEEK**MQTDRA NRFEYLLKQTELFAHFIQPAAQKTPTSPLKMKPGRPRIKKDEK**QNLL SVGDYR**HRRTEQEEDEELLTESSKATNVCTRFEDSPSYVKWGKLRD YQVRGLNWLISLYENGINGILADEMGLGKTLQTISLLGYMKHYRNI PGPHMVLVPKSTLHNWMSEFKRWVPTLRSVCLIGDKEQRAAFVRD VLLPGEWDVCVTSYEMLIKEKSVFKKFNWRYLVIDEAHRIKNEKSK LSEIVREFKTTNRLLLTGTPLQNNLHELWSLLNFLLPDVFNSADDFDS WFDTNNCLGDQKLVERLHMVLRPFLLRRIKADVEKSLPPKKEVK**IY VGLSK**MQREWYTRILMKDIDILNSAGKMDKMRLLNILMQLRKCCN HPYLFDGAEPGPPYTTDMHLVTNSGKMVVLDKLLPKLKEQGSRVL IFSQMTRVLDILEDYCMWRNYEYCRLDGQTPHDERQDSINAYNEP NSTK**FVFMLSTR**AGGLGINLATADVVILYDSDWNPQVDLQAMDR AHRIGQTKTVRVFRFITDNTVEERIVERAEMKLRLDSIVIQQGRLVD QNLNKIGKDEMLQMIRHGATHVFASKESEITDEDIDGILERGAKKT AEMNEKLSKMGESSLRNFTMDTESSVYNFEGEDYREKQK**IAFTEW IEPPKR**ERKANYAVDAYFREALRVSEPKAPKAPRPPKQPNVQDFQ FFPPRLFELLEKEILFYRKTIGYKVPRNPELPNAAQAQKEEQLKIDEA ESLNDEELEEKEKLLTQGFTNWNKRDFNQFIKANEKWGRDDIENIA REVEGKTPEEVIEYSAVFWERCNELQDIEKIMAQIERGEARIQRRISIK KALDTKIGRYKAPFHQLRISYGTNKGKNYTEEEDRFLICMLHKLGF DKENVYDELRQCIRNSPQFRFDWFLKSRTAMELQRRCNTLITLIER-ENMELEEKEKAEKKKRGPKPSTQKRKMDGAPDGRGRKKKLKL

### Statistics and reproducibility

All our experiments are reproducible, statistical methods are indicated in figure legends as well as sample sizes.

### Reporting summary

Further information on research design is available in the Nature Portfolio Reporting Summary linked to this article.

### Data availability

Source data for all figures except for proteomic data are available in the supplementary data file. Uncropped images of all the blots and gels used in the figures are provided in Supplementay Fig. 6. Proteomic data has been uploaded to PanoramaPublic web site as ProteomeXchange dataset PXD048865 (targeted parallel reaction monitoring of SMARCA5 peptides) and to PRIDE (Proteomics Identification Database) with project accession number: PXD049034 (SMARCA5 interactome data).

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

## Acknowledgements

We thank all members of the laboratory who participated in the discussions, in particular Dr. Lubomir Minarik. We would also like to thank Kristina Leblova for technical assistance. We acknowledge the Imaging Methods Core Facility at BIOCEV for their support with obtaining flow cytometry data presented in this paper, Laboratory of Mass Spectrometry at BIOCEV, where proteomic and mass spectrometric analyses had been done and Czech Center for Phenogenomics for mouse strain generation. Grant support: Agency of Czech Ministry of Health (NU22-05-00374, NU21-08-00312); Grant Agency of the Czech Republic (24-10435S, 24-10353S), and Charles University (UNCE/MED/016, Cooperatio, SVV260637); Next Generation EU, Programme EXCELES (LX22NPO5102). Czech Center for Phenogenomics was supported by the Czech Academy of Sciences RVO 68378050 and by the projects LM2018126 / LM2023036 Czech Center for Phenogenomics provided by Ministry of Education, Youth, and Sports of the Czech Republic (MEYS), CZ.02.1.01/0.0/0.0/16_013/0001789, and CZ.02.1.01/0.0/0.0/18_046/0015861 by MEYS & ESIF.

## Author contributions

Conceptualization TS, LC; methodology TZ, LC, JK, RS, TT, KP; investigation TT, JK, LC, KP, DN, ND, MH, LR; validation TT, JK, LC; formal analysis JK, LC, TT, KP; writing-original draft preparation TS, TT, JK; writing-review and editing TS, TT, LC, JK, ND, KP; visualization TT, JK; supervision TS, LC; project administration TS, LC; funding acquisition TS, LC. All authors have read and agreed to the published version of the manuscript.

## Competing interests

The authors declare no competing interests.
