## [Peer Review File · Communications Biology]

Reviewers' comments:

Reviewer #1 (Remarks to the Author):

The authors satisfactorily addressed all the comments from my initial review (reviewer #1) to their submission to Nature Communications. There are still some aspects that should be clarified to solidify the manuscript. Please see below my comments to the authors' answers to my previous review.

Major issues

1. and 2. Thanks for the further explanation about the similarities between human and mouse SMARCA5. However, the evidence provided in this new version (Supp. Fig. 1 A) only refers to amino acid sequencing and not functionality. As the authors acknowledge, post-transcriptional modifications may play a role in the role of SMARCA5 in different cell types. For this reason, the data the authors claim to have proving equal post-transcriptional processing in human and mouse for SMARCA5 should be included in this manuscript to add robustness to the model.

Also related to this point, as pure curiosity, if human SMARCA5 and mouse SMARCA5 are the same at all levels, why using the human protein to study murine HSC biology?

3. and 4. My apologies for the confusion. There was a typo in my comment. In comment #3, I meant Figure 2A of the original manuscript. Thanks for the explanation about the generation of the mouse model. However, the question about the expression levels from the S5tg remains open. Mice with one copy of this allele express different levels of human SMARCA5. For example, $S5^{fl/fl}S5^{tg}$ express twice human SMARCA5 than $S5^{del/fl}S5^{tg}$ in the thymus, while both have one $S5^{tg}$ copy (Figure 2A). Same for $S5^{fl/fl}S5^{tg/tg}$ and $S5^{del/fl}S5^{tg/tg}$. Does this mean that the deletion of the endogenous mouse allele affects the expression of the human one?

Reviewer #2 (Remarks to the Author):

Review Turkova et al, 2023 - Differential requirements for Smarca5 expression during hematopoietic stem cell commitment

This manuscript is a resubmission of an earlier manuscript previously submitted to another Nature journal. I note that the authors have responded to a good number of my comments and clarified some issues that were not clear for this reviewer. While the results are clearer, the main message of the manuscript remains the same, that Smarca5 is important in HSCs to enable proper formation of B and T cells, and more specifically in the maturation of DN cells to CD4 and CD8 T cells in the thymus, as well as in the formation of proB and pre-B cells in the bone marrow. Overall, the manuscript reads better and does provide new information on the role of Smarca5 in haematopoiesis – although a mechanistic insight is still largely absent.

I have a few issues that I'd like to see clarified prior to publication:

1- t-tests are used throughout the manuscript in the statistical comparisons (e.g. Fig 1d), comparing control to one of the genotypes. I believe that the appropriate statistics would be a one-way ANOVA with multiple comparisons. For example, again in Fig 1d, you want to compare the hCD2iCre $S5^{fl/fl}$, $S5^{tg}$ to wild type, but also to hCD2iCre $S5^{fl/fl}$ genotype. I suspect some of the conclusions about extent of rescues or severity of phenotypes might be slightly different once you've taken these into account.

For example, the stated rescue of pre-B cells by 1 allele of transgenic S5 is likely not statistically significant and would therefore affect your conclusions about the outcome of this experiment.

2- Non-significant comparisons are also important, so in instances where that is the conclusion, then

'ns' should be added to the relevant panel.

3- The transgenic Smarca5 allele is labelled as 'hypomorphic' at the beginning of the results section, but it's not clear anywhere why that is the case. Is the human smarca5 not fully functional when expressed in mouse? Or is it because from the data presented it is clear that the phenotype is not fully rescued? If the two copies of the transgenic alleles are 'able to almost fully rescue' the myphoid defect in the hCD2-iCRE induced Smarca5 KO, why would you label it as a hypomorph?

4- I find it puzzling why the levels of human smarca5 mRNA in the transgenics are similar to those of the wildtype mouse smarca5, yet protein levels are much lower.

5- The order of how the samples are presented (e.g. fig 2 C,D) with the single copy transgene last can lead to confusion – the idea is that the two copies of the transgenes are a better rescue, which would be helped by showing this particular sample last.

6- Line 206 – the 'significant' improvement of the rescue of definitive haematopoiesis seems to be a small effect at best – the use of the term significant in this context is a bit misleading

7- Line 212 – a rescue of myelocyte lineages by only 1 copy is also misleading as there is no difference in these cells irrespective of the smarca5 genotype.

8- This reviewer is still not convinced that the mass-spec experiment in the last figure adds significant value to this manuscript.

Reviewer #3 (Remarks to the Author):

The authors have addressed the reviewers' comments and improved the manuscript.

Reviewers' comments:

Reviewer #1 (Remarks to the Author):

The authors satisfactorily addressed all the comments from my initial review (reviewer #1) to their submission to Nature Communications. There are still some aspects that should be clarified to solidify the manuscript. Please see below my comments to the authors' answers to my previous review.

Major issues

1. and 2. Thanks for the further explanation about the similarities between human and mouse SMARCA5. However, the evidence provided in this new version (Supp. Fig. 1 A) only refers to amino acid sequencing and not functionality. As the authors acknowledge, post-transcriptional modifications may play a role in the role of SMARCA5 in different cell types. For this reason, the data the authors claim to have proving equal post-transcriptional processing in human and mouse for SMARCA5 should be included in this manuscript to add robustness to the model.

We apologize for the misunderstanding, but we do not have data demonstrating the similar post-transcriptional processing of the human and mouse SMARCA5. Our claim that the human and mouse SMARCA5 proteins behave the same in the mouse model and the phenotype described in our manuscript is not due to differences between the human and mouse variants of the protein is supported by the results of our other SMARCA5 transgenic model containing the mouse transgene. We do not intend to publish this model in this manuscript; it is an unrelated story, ready for publication on an unrelated topic. The mouse SMARCA5tg has a high similarity to human, however, inclusion of the data is completely beyond the scope of this manuscript. The mouse transgenic construct was inserted into the Rosa26 locus (similar to human S5tg) and uses a strong synthetic CAG promoter for expression instead of the endogenous Rosa26 promoter used for human S5tg described in our manuscript. Analysis of SMARCA5 expression in the murine S5tg model showed that reduced expression also occurs for this transgene, although the mouse transgene is expressed from a strong synthetic CAG promoter. We therefore conclude that the lower expression of the transgene is not due to a difference between mouse and human SMARCA5. Although the question of homology and tissue-specific action of the two variants is a very interesting topic, we believe that introducing a new mouse model into the manuscript would greatly expand its scale without yielding any significant or interesting results. The rationale for using our human S5tg instead of the mouse S5tg model was that human S5tg is expressed at a lower level due to a weaker promoter and is therefore a more appropriate model for studying defects in hematopoiesis that would not be observed in the mouse S5tg model.

Also related to this point, as pure curiosity, if human SMARCA5 and mouse SMARCA5 are the same at all levels, why using the human protein to study murine HSC biology?

The advantage of using a human S5tg model is that the sequences can be very well distinguished at the RNA level, which we have exploited in Fig. 1b and Fig. 2a. However, both transgenic models are suitable for research, it is more a question of which promoter they use and to what level the transgenic protein is expressed.

3. and 4. My apologies for the confusion. There was a typo in my comment. In comment #3, I meant Figure 2A of the original manuscript. Thanks for the explanation about the generation of the mouse model. However, the question about the expression levels from the S5tg remains open. Mice with one copy of this allele express different levels of human SMARCA5. For example, S5^{fl/fl}S5^{tg} express twice human SMARCA5 than S5^{del/fl}S5^{tg} in the thymus, while

both have one S5^{tg} copy (Figure 2A). Same for S5^{fl/fl}S5^{tg/tg} and S5^{del/fl}S5^{tg/tg}. Does this mean that the deletion of the endogenous mouse allele affects the expression of the human one?

We agree with the comment, however, the process of comparing expression at the transgenic mouse level is also interfered with by inter-individual variability and additional variability arising from the use of unsorted cell populations. The quantitative data can therefore be considered rather semi-quantitative. In particular, where the difference is not large or where standard deviations increase, it is very difficult to evaluate the data with confidence. The question posed by the opponent cannot therefore be answered unequivocally. Although the S5^{del/wt} S5^{tg(tg)} genotype expresses much less human S5 RNA than the S5^{fl/fl} S5^{tg(tg)} genotype, at the protein level we see that S5 expression is comparable, if not higher, given that S5^{del/wt} has only one endogenous allele and thus should theoretically have less protein. In interpreting our manuscript results, we were guided primarily by S5 protein level, which we believe is more important than RNA level. Unfortunately, at this time we cannot explain why the RNA level does not correspond to the protein level, nor why the S5^{del/wt} S5^{tg(tg)} genotype appears to express less human S5 RNA, although again this does not correspond to protein expression as discussed above. It is also important to mention that this RNA expression phenomenon can only be observed in the thymus. In our data from bone marrow, we observe a completely opposite trend, with the S5^{del/wt} S5^{tg(tg)} genotype expressing more human S5 RNA than the S5^{fl/fl} S5^{tg(tg)} genotype.

Reviewer #2 (Remarks to the Author):

Review Turkova et al, 2023 - Differential requirements for Smarca5 expression during hematopoietic stem cell commitment

This manuscript is a resubmission of an earlier manuscript previously submitted to another Nature journal. I note that the authors have responded to a good number of my comments and clarified some issues that were not clear for this reviewer. While the results are clearer, the main message of the manuscript remains the same, that Smarca5 is important in HSCs to enable proper formation of B and T cells, and more specifically in the maturation of DN cells to CD4 and CD8 T cells in the thymus, as well as in the formation of proB and pre-B cells in the bone marrow. Overall, the manuscript reads better and does provide new information on the role of Smarca5 in haematopoiesis – although a mechanistic insight is still largely absent.

I have a few issues that I'd like to see clarified *prior to publication*:

1- t-tests are used throughout the manuscript in the statistical comparisons (e.g. Fig 1d), comparing control to one of the genotypes. I believe that the appropriate statistics would be a one-way ANOVA with multiple comparisons. For example, again in Fig 1d, you want to compare the hCD2iCre S5^{fl/fl}, S5^{tg} to wild type, but also to hCD2iCre S5^{fl/fl} genotype. I suspect some of the conclusions about extent of rescues or severity of phenotypes might be slightly different once you've taken these into account.

For example, the stated rescue of pre-B cells by 1 allele of transgenic S5 is likely not statistically significant and would therefore affect your conclusions about the outcome of this experiment.

We agree with this comment and have changed the statistical analysis in Figures 1, 2 and 3 from a t-test to a one-way ANOVA, which has been newly indicated in the figure legend. However, the new type of analysis changed the significances only slightly and did not change our main conclusions. We are grateful for your comment on the rescue of pre-B cells in hCD2iCre mice; there was indeed an error and the sentence in the first part of results has been changed accordingly: The single S5^{tg} allele was unable to increase the number of pre-B cells, but in the presence of both S5^{tg} alleles we observed almost complete rescue (Fig. 1h). And we also made adjustments in this sentence in same section of the results: This

rescue was incomplete for DN1 and DN4 thymocytes, consistent with the reduced cellularity of the $S5^{tg}$ thymi (Fig. 1f, 1g). And we also made some adjustments in section 3 of the results because with the new statistical method, the accumulation of $CD34^{+}FLT3^{-}$ cells is statistically significant only in single $S5^{tg}$ animals. Changes were made in this part of the text: The $CD34^{+}FLT3^{-}$ LSK cell population was accumulated in single $S5^{tg}$ mice compared to controls, representing up to a ten-fold increase in $Vav1Cre S5^{fl/del}S5^{tg}$ mice (Suppl. Fig. 3b, 3f). However, this net increase almost entirely consisted of MPP3 ($CD48^{+}CD150^{-}$), which can contribute to both myeloid and lymphoid lineages, and also of myeloid-committed MPP2 ($CD48^{+}CD150^{+}$) (Fig. 3e, S3c).

2- Non-significant comparisons are also important, so in instances where that is the conclusion, then 'ns' should be added to the relevant panel.

We agree that it is useful to include as much information as possible in the image but inserting "ns" would be detrimental to clarity. We believe that this would make the graphs busier and therefore harder to read, so we have opted for the option to show only cases where there have been significant changes in the graphs, which are expressed by the appropriate number of asterisks (*). Where there are no asterisks, no significant changes have occurred, which we have indicated in the figure legends (no asterisks = not significant).

3- The transgenic *Smarca5* allele is labelled as 'hypomorphic' at the beginning of the results section, but it's not clear anywhere why that is the case. Is the human *smarca5* not fully functional when expressed in mouse? Or is it because from the data presented it is clear that the phenotype is not fully rescued? If the two copies of the transgenic alleles are 'able to almost fully rescue' the myeloid defect in the hCD2-iCRE induced *Smarca5* KO, why would you label it as a hypomorph?

This is a topic discussed repeatedly even within our group, we labelled our allele as a hypomorph because its expression on protein level is significantly lower as compared to the endogenous protein. Although transgenic *SMARCA5* is functional, it cannot fully replace endogenous *SMARCA5* in the populations that were most important to us in our study (hematopoietic stem cells and progenitors) due to its lower protein levels. We decided to add an explanation in the introduction of the manuscript, where the hypomorphic allele is mentioned for the first time: To prevent this phenomenon and to study in detail the role of *SMARCA5* in early definitive hematopoiesis, we decided to create a transgenic model with a hypomorphic expression (lower expression on protein level) of *SMARCA5* in the context of its endogenous gene deletion.

4- I find it puzzling why the levels of human *smarca5* mRNA in the transgenics are similar to those of the wildtype mouse *smarca5*, yet protein levels are much lower.

We agree that this is an important point that we have decided to highlight in the discussion. We suggest that there is a regulatory mechanism for *SMARCA5* at the protein level because our results from RNA expression do not match those from protein expression. This regulatory mechanism is the subject of our next study, which we are currently working on. When interpreting the results of our manuscript, we were mainly guided by the *S5* protein levels, which in our view are more important than the RNA levels. Unfortunately, at the moment we cannot explain why RNA level does not correspond to protein level. This text was added to the discussion: Interestingly, we observed a discrepancy between *SMARCA5* transgene expression at the RNA and protein levels, with the transgene being able to reach endogenous *Smarca5* RNA level but not the protein level suggesting a regulatory mechanism for *SMARCA5* at the protein level requiring further investigation.

5- The order of how the samples are presented (e.g. fig 2 C,D) with the single copy

transgene last can lead to confusion – the idea is that the two copies of the transgenes are a better rescue, which would be helped by showing this particular sample last.

We apologize for the disagreement in this regard, but we still think that our order of samples is clearer. We chose to present the samples in an order that is corresponding to the severity of the phenotype, with the WT control first, followed by mice with deletion of only one allele of endogenous S5 that also express transgenic S5, and then mice with deletion of endogenous S5 that express only transgenic S5. Based on the severity of the phenotype, mice with two S5tg alleles are always placed before mice with one S5tg allele. Another option would be to arrange the genotypes by rescue level, as you suggest, but we found the ordering by phenotype severity clearer, given the interpretation of the results.

6- Line 206 – the ‘significant’ improvement of the rescue of definitive haematopoiesis seems to be a small effect at best – the use of the term significant in this context is a bit misleading

Although the better rescue effect of two alleles compared to one is statistically significant, as shown in the figure, we agree that this term is misleading, so we decided to remove the word ‘significantly’ from the text.

7- Line 212 – a rescue of myelocyte lineages by only 1 copy is also misleading as there is no difference in these cells irrespective of the smarca5 genotype.

We are sorry for this confusion, we wanted to state that myelocytes are rescued in all genotypes, even in the genotype with a single copy of the transgene. We changed the sentence from: which were completely rescued with a single allele of $S5^{tg}$, to: which were completely rescued ‘even by just a’ single allele of $S5^{tg}$.

8- This reviewer is still not convinced that the mass-spec experiment in the last figure adds significant value to this manuscript.

Thank you for your comment, however, this experiment provides highly interesting data and also quite unique and had always had a positive response at presentations, so we would like to leave this part as it is. This experiment was also positively received by other reviewers.

Reviewer #3 (Remarks to the Author):

The authors have addressed the reviewers' comments and improved the manuscript. Thank you for reading and reviewing our manuscript.

We also made small adjustments in the grant support section of the acknowledgements: Grant support: Agency of Czech Ministry of Health (NU22-05-00374, NU21-08-00312); Grant Agency of the Czech Republic (24-10435S, 24-10353S), and Charles University (UNCE/MED/016, Cooperatio, SVV260637); Next Generation EU, Programme EXCELES (LX22NPO5102). Czech Center for Phenogenomics was supported by the Czech Academy of Sciences RVO 68378050 and by the projects LM2018126 / LM2023036 Czech Centre for Phenogenomics provided by Ministry of Education, Youth, and Sports of the Czech Republic (MEYS), CZ.02.1.01/0.0/0.0/16_013/0001789, and CZ.02.1.01/0.0/0.0/18_046/0015861 by MEYS & ESIF.

REVIEWERS' COMMENTS:

Reviewer #1 (Remarks to the Author):

Thanks to the authors for addressing my remaining comments.

This reviewer is still concerned about the inconsistencies in the expression levels of human SMARCA5. The authors argue that although they do not have any explanation for the discrepancies in RNA expression, they show equal or increased SMARCA5 expression in the $S5^{tg/tg}$ samples compared to $S5^{tg}$, by protein. However, $S5^{fl/fl}$, $S5^{fl/fl}S5^{tg}$ and $S5^{fl/fl}S5^{tg}$ cells show very similar levels of SMARCA5 expression (Supp Fig 2b). This suggests that the human SMARCA5 allele expression is affected by the presence of the murine allele (or protein). The authors also mentioned that the data showed should be considered semi-quantitative due to biological and technical reasons ("inter-individual variability and additional variability arising from the use of unsorted cell populations"). These data are important to validate the experimental model and the lack of clarity on it reduces the robustness of the study.

Other than this, I do not have any further comments.

Reviewer #2 (Remarks to the Author):

I thank the authors for their diligence in responding to my comments and arguing their case where we disagreed; I have no further comments to make and happy for the manuscript to be accepted.

Reviewer #1 (Remarks to the Author):

Thanks to the authors for addressing my remaining comments.

This reviewer is still concerned about the inconsistencies in the expression levels of human SMARCA5. The authors argue that although they do not have any explanation for the discrepancies in RNA expression, they show equal or increased SMARCA5 expression in the $S5^{tg/tg}$ samples compared to $S5^{tg}$, by protein. However, $S5^{fl/fl}$, $S5^{fl/fl}S5^{tg}$ and $S5^{fl/fl}S5^{tg}$ cells show very similar levels of SMARCA5 expression (Supp Fig 2b). This suggests that the human SMARCA5 allele expression is affected by the presence of the murine allele (or protein). The authors also mentioned that the data showed should be considered semi-quantitative due to biological and technical reasons (“inter-individual variability and additional variability arising from the use of unsorted cell populations”). These data are important to validate the experimental model and the lack of clarity on it reduces the robustness of the study.

Other than this, I do not have any further comments.

Thank you for your comment and also for your efforts to interpret our data correctly. Your feedback has led us to a much more detailed analysis of the source data by the whole team. Specifically, we found that one data set was not correctly converted into graphical form, and this led to a larger variability in the resulting graphs. Specifically, for the $S5^{del5/wt} S5^{tg(tg)}$ genotypes, we mistakenly replaced the 2nd derivative max number of cycles from the qPCR with the fit points (which we used for other samples). Because the 2nd derivative max cycle count is always higher than the cycle count of the fit point, this caused us to have less total S5 mRNA in our plots in the $S5^{del5/wt} S5^{tg(tg)}$ genotype. We have revised and corrected Figure 2A, and it can now be seen that there is indeed more human S5tg mRNA in the $S5^{del5/wt} S5^{tg(tg)}$ genotypes than in the $S5^{fl/fl} S5^{tg(tg)}$ genotypes. The newly corrected data refute the hypothesis that deletion of the endogenous allele negatively affects expression of the transgenic allele at the mRNA level. We apologize for the confusion in communication but at the same time are very grateful that we identified this minor error in our mRNA data and subsequently corrected the figure. However, this did not require any changes to the text or modify our earlier conclusions.

Updated figure:

Fig2(a) Copy number quantification of mouse and human Smarca5 mRNA by qRT-PCR (AVG±SD, n=3) in thymi and BM of 2-month-old mice of the indicated genotypes after normalization to Gapdh and Hprt expression.

Additional changes:

1. We have changed the format of Figures 1b, 2a and 3i so that they now show individual values.
2. We added these two sentences to the end of the Materials and Methods section, subsection Mouse transgenesis: We have complied with all relevant ethical regulations for animal use. In all experiments, we used mice of both sexes, age is indicated in figure legends.
3. We added Statistics and Reproducibility subsection to the Materials and Methods section: All our experiments are reproducible; statistical methods are provided in the figure legends, as are sample sizes and repetition rates where applicable.
4. Size markers were added to Figures 1c, 2b, S1d and S2b.
5. Uncropped and unedited blot images were added to Supplemental Figures and this sentence was added to Materials and Methods section, subsection Immunoblotting: Uncropped and unedited blot images are available in Supplementary Figure 6.
6. This statement was added to the end of the legend of each figure and supplementary figure: the error bars represent standard deviation.
7. This data availability statement was added to the end of the manuscript: Source data for all figures except for proteomic data are available in the supplementary material to this manuscript. Proteomic data has been uploaded to PanoramaPublic web site as ProteomeXchange dataset PXD048865 (targeted parallel reaction monitoring of SMARCA5 peptides) and to PRIDE (Proteomics Identification Database) with project number: ...* (SMARCA5 interactome data).
* We are currently uploading this dataset to the PRIDE website and will provide the project number within 7 days.